# Lineage-informative microhaplotypes for recurrence classification and spatio-temporal surveillance of *Plasmodium vivax* malaria parasites

Challenges in classifying recurrent *Plasmodium vivax* infections constrain surveillance of antimalarial efficacy and transmission. Recurrent infections may arise from activation of dormant liver stages (relapse), blood-stage treatment failure (recrudescence) or reinfection. Molecular inference of familial relatedness (identity-by-descent or IBD) can help resolve the probable origin of recurrences. As whole genome sequencing of *P. vivax* remains challenging, targeted genotyping methods are needed for scalability. We describe a *P. vivax* marker discovery framework to identify and select panels of microhaplotypes (multi-allelic markers within small, amplifiable segments of the genome) that can accurately capture IBD. We evaluate panels of 50–250 microhaplotypes discovered in a global set of 615 *P. vivax* genomes. A candidate global 100-microhaplotype panel exhibits high marker diversity in the Asia-Pacific, Latin America and horn of Africa (median $H_E = 0.70–0.81$) and identifies 89% of the polyclonal infections detected with genome-wide datasets. Data simulations reveal lower error in estimating pairwise IBD using microhaplotypes relative to traditional biallelic SNP barcodes. The candidate global panel also exhibits high accuracy in predicting geographic origin and captures local infection outbreak and bottlenecking events. Our framework is open-source enabling customised microhaplotype discovery and selection, with potential for porting to other species or data resources.

The malaria parasite *Plasmodium vivax* is a major public health threat affecting the poorest and most vulnerable populations in more than 49 endemic countries[1]. Over the past decade, enhanced malaria control efforts in areas outside of sub-Saharan Africa have achieved a marked decline in *P. falciparum* infections, but a relative rise in the proportion of *P. vivax* cases[1]. Several biological attributes of *P. vivax* render this species more resilient to interventions than *P. falciparum*[2]. *P. vivax* forms dormant liver stage parasites (hypnozoites) that can reactivate weeks to months after initial inoculation causing recurrent episodes of malaria (relapses). A single mosquito inoculation can cause multiple relapses, sustaining transmission over extended periods. Relapses are thought to cause over 60% of clinical cases[3]. Knowledge of the biology and epidemiology of *P. vivax* relapse, including the underlying reactivation mechanism(s) and host and parasite determinants, are vital for informing public health strategies to combat this parasite, but our understanding of these processes is limited[4]. A major obstacle to our increased understanding has been the difficulty of classifying recurrent *P. vivax* infections accurately. These recurrences can arise from reinfection (new mosquito inoculations), recrudescence (blood-stage treatment

✉e-mail: Sarah.Auburn@Menzies.edu.au

failures), or relapse (reactivation of dormant liver stages). Discriminating between these causes is challenging[1].

Accurate methods to disentangle relapse from reinfection and recrudescence events are critical to improving our understanding of the therapeutic efficacy of current treatment regimens for *P. vivax*. Accumulating reports suggest that chloroquine, the most widely used drug for treating the *P. vivax* blood stages, is failing in several endemic regions, but recrudescence (drug failure) can be confused with relapse, confounding efficacy studies[5]. Accurate diagnosis of the cause of recurrent infection is also essential to clinical studies of hypnozoiticidal (anti-relapse) regimens (primaquine and tafenoquine) as this relies on distinguishing relapse from reinfection[6,7]. Reinfection dilutes observed differences in recurrence between hypnozoiticidal interventions and biases treatment effect estimates towards interventions which provide longer post-treatment prophylaxis[8].

The ability to discriminate between recurrent *P. falciparum* infections deriving from reinfections (which are likely genetically heterologous to the initial infection given sufficient population diversity) and recrudescences (which contain homologous parasites) using genotyping at a handful of polymorphic markers, represented a crucial advance for *P. falciparum* clinical research[9]. It allowed therapeutic efficacy studies to be conducted in endemic areas without the need for concomitant detailed epidemiological assessment. However, the situation is more complex for *P. vivax* malaria since relapsing parasites can be genetically homologous or heterologous to the initial infection[10,11]. However, recent genomic studies have revealed that a proportion of paired *P. vivax* isolates (from acute and recurrent infections) that would be classified as heterologous using traditional genotyping approaches share homology in large segments of the genome, inferring familial relatedness[12–14]. In mosquito hosts capturing blood meals with mixtures of parasite genomes, the obligate meiotic stage will generate meiotic recombinants, producing sporozoites that share parents. Every natural infection deriving from more than one sporozoite may comprise mixtures of meiotic recombinants. Pairs of infections with evidence of recent identity by descent (IBD) consistent with close relationships such as siblings or half-siblings (as much as ≥50% and ≥25% IBD respectively), are more likely to have derived from the same mosquito inoculation than from different inoculations and are, therefore, more likely to reflect relapse than reinfection events. IBD therefore has significant potential to enable more accurate classification of recurrent *P. vivax* infections. IBD-based measures would also allow finer resolution of the spatial connectivity between *P. vivax* populations than is possible using classical methods such as the fixation index or phylogenetic approaches[15]. With appropriate genetic data, IBD can be used to characterise major transmission networks (foci) for targeted intervention, or the risks of infection spread between communities and across borders.

While genomic data provides the greatest information to estimate IBD, *P. vivax* patient isolates often have low parasite densities which currently precludes whole genome sequencing at large scale, even when using selective whole genome amplification methods[16]. Restricting analyses to infections with high parasite densities is not ideal as these are atypical of *P. vivax* infection and hence may not be representative of the true diversity in patient infections. High-throughput genotyping offers a more operationally feasible approach that can be applied to low-volume samples such as dried blood spots and would be more readily implemented in surveillance frameworks in malaria-endemic countries[17–20]. However, selecting parsimonious marker sets that can capture genome wide IBD effectively is challenging[21]. To date, genotyping methods for *P. vivax* have relied on either capillary sequencing of microsatellite markers or next generation sequencing (NGS) of Single Nucleotide Polymorphisms (SNPs)[2,22]. In a recent *P. falciparum* study, targeted NGS of short regions comprising multiple highly variable SNPs (microhaplotypes) provided a simpler, cheaper, and higher-throughput approach than microsatellite typing, with substantially higher resolution of individual clones than with single SNPs[23].

In this work, we establish a framework for selecting and evaluating the effectiveness of universal *P. vivax* IBD barcodes that can be used to improve the interpretation of therapeutic evaluations of hypnozoiticidal and schizonticidal antimalarial drugs, elucidate the biology and epidemiology of *P. vivax* relapses, and provide NMCPs and other agencies with actionable information on parasite transmission within and across national borders. Using 615 publicly available *P. vivax* genomes, we identify panels of microhaplotype markers that can estimate IBD across pairs of infections in diverse parasite populations. In silico analyses confirm that *P. vivax* microhaplotypes can provide much higher resolution to estimate relatedness consistent with close relationships compared to biallelic SNP barcodes that are currently used. We demonstrate that carefully selected panels for IBD characterisation can not only resolve *P. vivax* relatedness relationships within patients, but also have broad utility for spatiotemporal use cases.

## Results

### Microhaplotype marker discovery framework and selection

We created a microhaplotype marker discovery framework to identify and select panels of microhaplotypes with high accuracy in capturing IBD and potential to be incorporated into amplicon-based sequencing assays. The primary marker selection criteria were: i) high diversity and preferably multiallelic; and ii) even spacing throughout the genome. Variable panel sizes can be selected but a minimum of 100 microhaplotypes is recommended. The framework has been built as a series of Jupyter notebooks in Python and was implemented on the MalariaGEN Pv4 dataset. The notebooks are directly connected to the Pv4 data resource on Google Colab enabling users to discover and analyse their own marker panels without downloading data or files. The discovery framework walks users through the entire microhaplotype selection process and is customisable to different sample sets, marker properties and panel sizes with embedded exploratory benchmarking analysis for marker informativeness https://svsiegel.github.io/vivax-mhaps/, https://doi.org/10.5281/zenodo.12622789.

The first step of the discovery framework (notebook 1: svsiegel/vivax-mhaps) entails sample quality filtering; an initial set of 1,816 isolates in the MalariaGEN Pv4 dataset were filtered to 615 high quality, likely monoclonal samples (Fig. 1a, Supplementary Data 1)[24]. In addition to country classifications, the MalariaGEN Pv4 samples were assigned regional classifications based on genetic clustering patterns[24]. Although most parasites originated from the Asia-Pacific region, the dataset exhibited global geographic representation, spanning 17 countries, with all regional populations represented by at least 30 isolates (range n = 31–151) (Table 1, Supplementary Fig. 1). The filtered sample subset was used to select Single Nucleotide Polymorphisms (SNPs) and identify candidate microhaplotypes within 200 bp regions of the *P. vivax* genome (Fig. 1b) with high diversity in all geographic regions selected.

A total of 13,084 candidate high-quality biallelic SNPs with global high minor allele frequency (MAF ≥ 0.1) and low genotype failure rate (missingness <0.1) were identified in core, coding regions of the genome to minimise potential primer design challenges (Fig. 1a). A windowed scan was conducted in partially overlapping windows across all identified variants of interest (13,498 windows found). As illustrated in Fig. 1c, windows were relatively uniformly distributed across the genome. We then filtered further by heterozygosity to be above the theoretical maximum for a single biallelic SNP of 0.5 and having 3 or more SNPs within the window. Together, this framework identified a total of 3,830 microhaplotype candidate heterozygome windows within the core genome.

The first step in panel selection (notebook 2: svsiegel/vivax-mhaps) divides the total length of the 14 chromosomes by the number

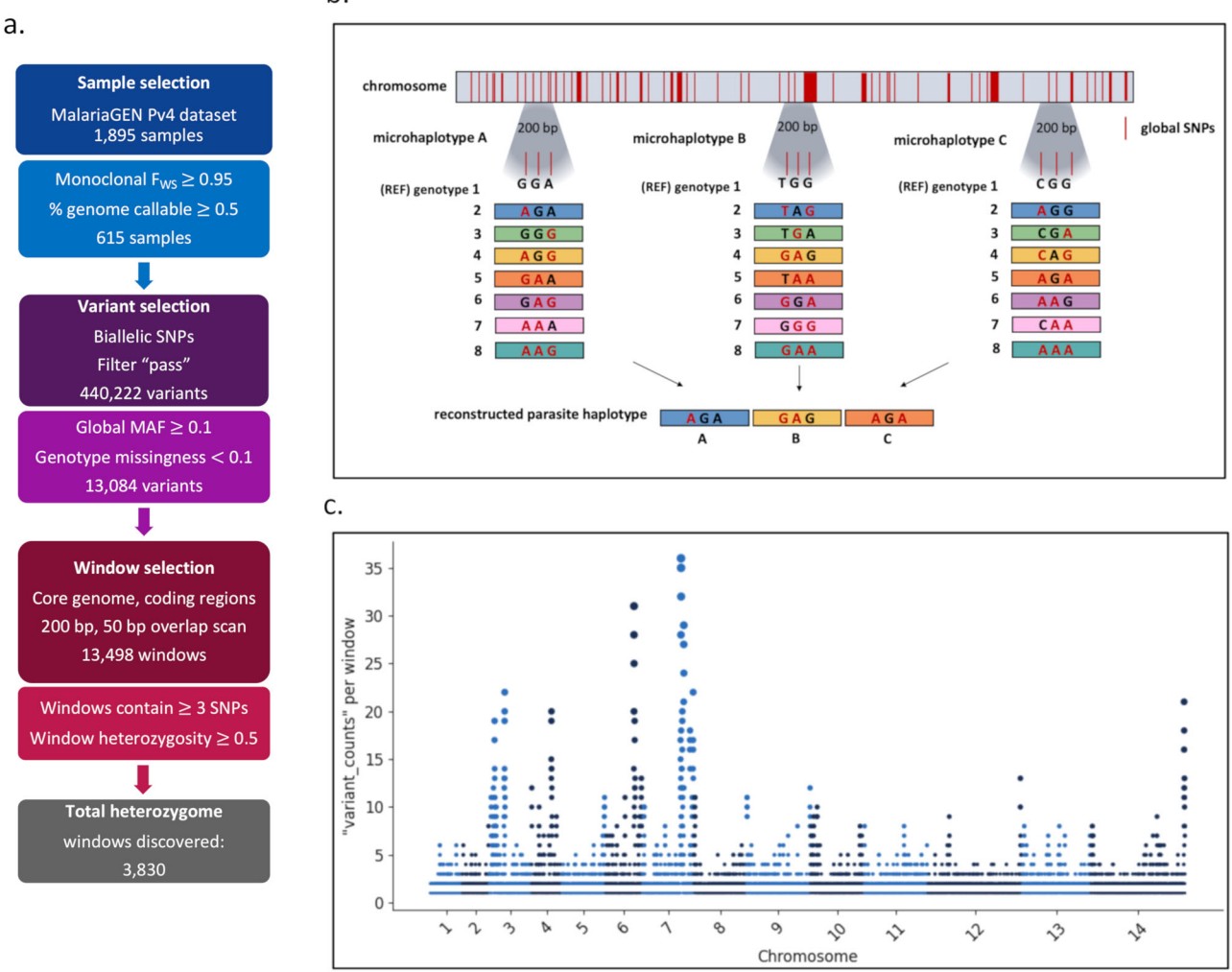

**Fig. 1 | Microhaplotype discovery pipeline.** Panel (**a**) provides an overview of the marker selection process. Criteria for selecting samples, variants, and windows for potential microhaplotype candidate windows result in a total of 5460 windows (200 bp). The MalariaGEN Pv4 dataset was filtered to use only high-quality monoclonal samples ($F$ws ≥ 0.95) that had at least 50% of the core genome positions callable. SNP variants from this sample subset were then identified as biallelic, have low genotype missingness (<0.1), had high global minor allele frequencies (MAF ≥ 0.1), and FILTER = PASS in the MalariaGEN dataset, resulting in 13,084 total SNPs. The core genome was then scanned in coding regions for all 200 bp windows in which > 3 of the identified variants were found and filtered for high diversity (global heterozygosity ≥0.5). Panel (**b**) provides a schematic representation of microhaplotypes with 3 SNPs. Microhaplotypes leverage SNP information content in small-windowed regions of the genome to provide a high-resolution reconstruction of the parasite genome. Three high-diversity SNPs in a single microhaplotype can have as many as 8 distinct combinations of alleles which, when combined with 100 or more microhaplotypes across the genome, results in high discriminatory power to characterise relatedness. **c** Manhattan plot of the *P. vivax* windows identified across the genome, showing the chromosomal distribution of all windows identified in the global set of high-quality, independent monoclonal infections with at least 1 SNP. Each point is an identified window, with the size increasing as the number of SNPs within the window increases. Potential microhaplotype regions are well distributed across the 14 chromosomes. The microhaplotypes with the highest SNP densities tend to be located at the ends of the chromosomes. Note, microhaplotypes were selected only from the core regions of the genome i.e., excluding highly diverse telomeric, sub-telomeric and centromere regions where sequence reads could not be mapped accurately (see Pv4 data resource for further details on core regions).

of desired markers (default 100) to assign the proportional number of markers needed per chromosome. The framework then provides user-defined options to generate panels prioritised for either uniform spacing (see notebook 2 evenly spaced algorithm), or for heterozygosity (see notebook 2 greedy algorithm) (Fig. 2a). Candidate panels can also be manually refined to balance both heterozygosity and marker spacing (Fig. 2b).

For evaluation of random versus high-diversity optimisation, we created two comparative 100-microhaplotype panels, both with 3–10 SNPs, global heterozygosity >0.6, and final marker selection curated manually. The exemplar panel contained well-spaced, high heterozygosity markers (high-diversity panel), while the comparator panel had well-spaced markers without heterozygosity optimisation (random panel) (Fig. 2c). The rationale for selecting 3–10 SNPs was based

on diminishing returns observed above 10 SNPs (see exploratory analyses - notebook 2) and retaining a computationally manageable number of allele combinations ($2^{10}$ allele combinations possible = 1024).

The high-diversity and random 100-microhaplotype panels were also benchmarked against a previously developed 42-SNP biallelic panel (Broad barcode) that has been widely used by the vivax community[25]. Four SNPs within the Broad barcode had to be excluded from analyses because of high genotype failure rates or multi-allelic status in the Pv4 dataset, leaving a 38-SNP Broad barcode for evaluation (Fig. 2c).

For evaluation of panel size, we also created five panels with 50, 100, 150, 200 and 250 microhaplotypes using the greedy algorithm with roughly even spacing. Panels were chosen to contain 3–10

**Table 1 | Geographic distribution of the sample set**

| Region | MalariaGEN Pv4 dataset | | | Independent validation *P. vivax* dataset | |
|---|---|---|---|---|---|
| | No. isolates sequenced | No. high-quality independent monoclonal isolates | No. high-quality paired isolates | No. isolates sequenced* | No. high-quality independent monoclonal isolates |
| Africa (AF) | 47 | 34 | 0 | 48 | 47 |
| East Southeast Asia (ESEA) | 262 | 151 | 6 | 494 | 62 |
| Maritime Southeast Asia (MSEA) | 76 | 59 | 4 | 1 | 1 |
| Oceania (OCE) | 205 | 120 | 2 | 1 | 1 |
| South America (SAM) | 159 | 135 | 0 | 203 | 92 |
| West Asia (WAS) | 46 | 31 | 0 | 88 | 87 |
| West Southeast Asia (WSEA) | 127 | 85 | 2 | 1 | 1 |
| Total | 922 | 615 | 14 | 836 | 291 |

The countries represented within each regional group of the MalariaGEN Pv4 dataset are Ethiopia (AF), Cambodia, Thailand and Vietnam (ESEA), Malaysia (MSEA), Indonesia and Papua New Guinea (OCE), Brazil, Colombia, El Salvador, Mexico, Nicaragua and Peru (SAM), Afghanistan, India and Iran (WAS), and Myanmar and Thailand (WSEA). The Thai samples in ESEA and in WSEA derive from provinces east and west, respectively, of a previously described malaria-free corridor that runs through the centre of Thailand[44,55]. The countries represented in the independent (non-Pv4) *P. vivax* dataset are Eritrea, Ethiopia, Madagascar, Mauritania, Sudan and Uganda (AF), Cambodia and China (ESEA), The Philippines (MSEA), Papua New Guinea (OCE), Brazil, Colombia, Guyana, Panama and El Salvador (SAM), Afghanistan, India and Pakistan (WAS) and Bangladesh (WSEA).

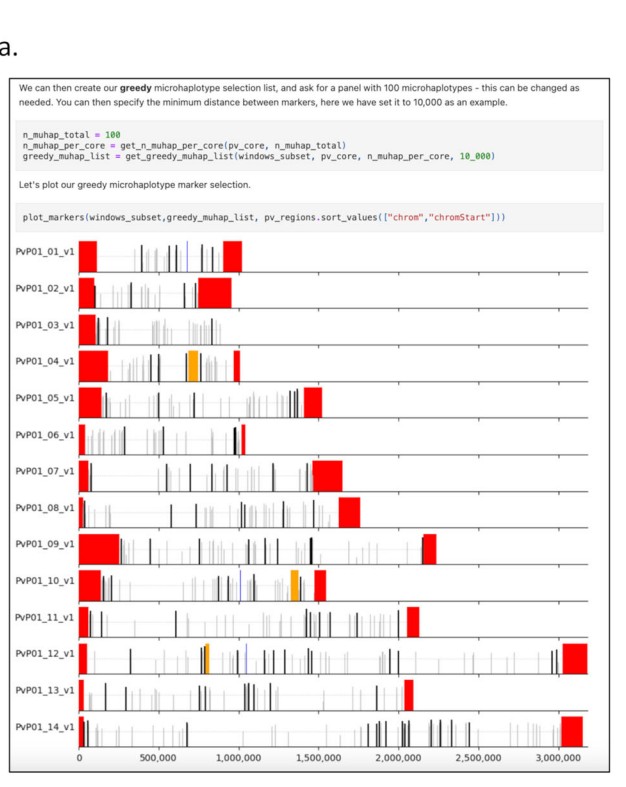

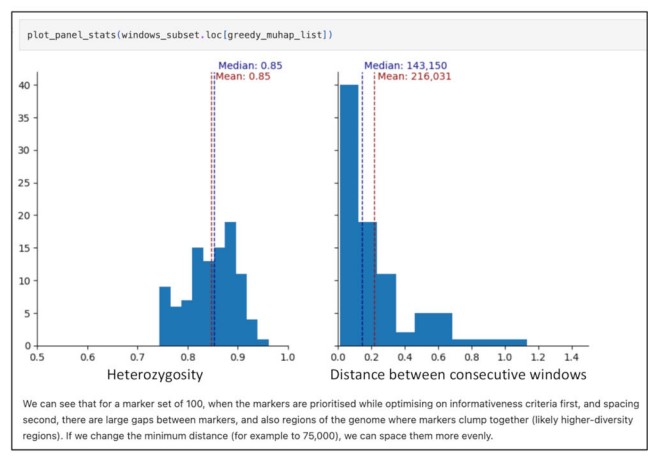

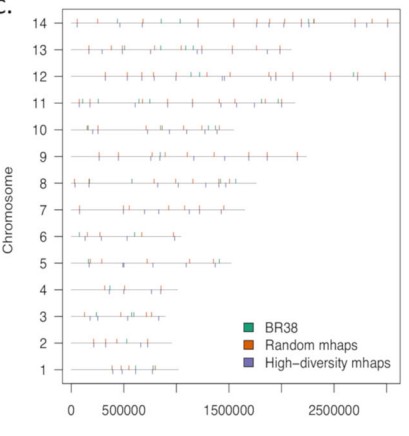

**Fig. 2 | Marker property outputs within the discovery pipeline.** Panel (**a**) illustrates an example marker selection plot output from the marker selection framework. Panel (**b**) illustrates examples of marker heterozygosity and distance distributions output from the marker selection framework. Panel (**c**) illustrates the chromosome distribution of three panels (two of which were selected within the framework) that were evaluated for their capacity to reconstruct parasite relatedness. Two new microhaplotype panels were created from the microhaplotype discovery pipeline, named "Random mhaps" plotted in orange, and "High-diversity mhaps" plotted in purple. These two panels were selected to have 100

microhaplotype markers using windows that were well-spaced and had between 3–10 SNPS, with even distribution across all 14 chromosomes and a minimum diversity with heterozygosity ≥0.65. The markers for the Random mhaps panel were selected randomly without optimisation, while the High-diversity mhaps panel were optimised to have the highest heterozygosity possible in each region. These two high-resolution panels were compared to the commonly used biallelic 42-SNP panel, named "BR38" in green. Only 38 markers of this panel are considered informative and included in this representation.

candidate SNPs and were optimised for highest heterozygosity. Details on the coordinates and SNPs covered by all microhaplotype panels are provided in Supplementary Data 2. A full description of the optimal selection criteria identified and demonstrated in this manuscript can be found in notebooks 1 and 2–svsiegel/vivax-mhaps and directly executable on Github pages https://svsiegel.github.io/vivax-mhaps/.

## Evaluation of randomly selected versus high-diversity microhaplotype panels on IBD estimation using simulated data

In each geographic region, the 95% confidence intervals (CIs) around the estimate $\hat{r}$ were calculated as a measure of the informativeness of a given marker panel (Fig. 3). In all geographic regions and for all marker panels, the CI intervals were shortest when $r = 1$ (identical infections), closely followed by $r = 0$ (strangers), and highest when $r = 0.5$ (siblings) indicating lowest accuracy in predicting sibling relationships. In all geographic regions, the CI intervals were smallest around the estimate $\hat{r}$ for the high-diversity microhaplotype panel followed by the random microhaplotype panel, and the 38-SNP Broad barcode (Fig. 3). At the high-diversity panel, all (100%) of the estimates of $r = 0.5$ had 95% CIs between 0.1 and 0.8. Whilst the Random-SNP microhaplotype panel followed similar trends to the high-diversity panel in each geographic region, the BR38 barcode displayed particularly large CI intervals relative to the microhaplotype panels in Maritime Southeast Asia and Oceania. The same trends in marker panel performance were observed using the root mean squared error (RMSE) of the estimates of $\hat{r}$ compared to the data-generating $r$ (Supplementary Fig. 2). In all regions, the high-diversity microhaplotype panel exhibited consistently lower RMSE values than the random microhaplotype and the 38-SNP Broad barcode. Importantly, the high-diversity panel was the only panel able to consistently keep RMSE values below 0.1 in all subpopulations (even when $r = 0.5$), which shows that selection with highest heterozygosity microhaplotypes outperformed the random microhaplotype panel in relatedness estimates. These results indicate that panels of ~100 uniformly spaced, high heterozygosity microhaplotypes provide more accurate estimation of IBD than the current single SNP panel used for *P. vivax* infections; this advantage may be more pronounced in certain geographic regions, possibly reflecting the geographic representation informing SNP selection. All downstream analyses focused on the 100 marker panels for evaluation of capturing IBD, including the high-diversity and random microhaplotype panels for comparison, with the high-diversity panel serving as a hypothetical exemplar panel for IBD use cases.

## Evaluation of an exemplar high-diversity 100 microhaplotype panel on IBD estimation using real data

In addition to the simulated data, the efficacy of the high-diversity 100 microhaplotype panel in estimating IBD between infections was evaluated using real data from the independent, high-quality monoclonal infections in Pv4. Pairwise measures of IBD were determined for the genome-wide SNPs and for the microhaplotype SNPs using *hmmIBD*. In all geographic regions, the microhaplotype-based estimates of pairwise IBD demonstrated a significant positive correlation with the genome-wide estimates (all $P < 0.05$, Spearman's rho statistic using a paired test, Supplementary Fig. 3).

Further evaluation using real data was undertaken on sequenced pairs of primary and recurrent *P. vivax* infections in Pv4. These samples came from a range of clinical trials, as well as from returning travellers[26–29]. A total of 14 infection pairs satisfied the criteria for selection which required monoclonal infections with high-quality genomic data at both time points. The infection pairs originated from sites in the Asia-Pacific and exhibited a range of durations between the primary and recurrent episode (Table 2). Genome-wide IBD measures using *hmmIBD* revealed that 11 infection pairs were clones (IBD ≥ 0.95) reflecting potential recrudescence or relapse events, whilst 1 pair was distant relatives (0.05 ≤ IBD < 0.25) and 2 pairs were strangers (IBD <

0.05). The distant relatives and strangers reflect potential reinfection or relapse events. When the data were restricted to the 100-microhaplotype markers, the resulting *hmmIBD*-based estimates correlated highly with the genome-wide measures of IBD (rho=0.790, Spearman's rank correlation). One pair of infections (PJ0167-C and PJ0166-C) that were classified as strangers with genomic data (IBD = 0.048) were classified as distant relatives (IBD = 0.156) with the microhaplotypes, but the other 13 pairs had concordant recurrence classifications.

## Microhaplotype panels can effectively capture diversity and differentiation

Further evaluation of the high-diversity 100 microhaplotype panel was conducted to assess the utility of such panels in capturing key population genetic features. The genetic diversity of the panel was assessed for isolates from each of the 7 geographic regions using measures of heterozygosity and effective cardinality. Heterozygosities across the microhaplotypes varied, but overall diversity was high in all geographic regions. The lowest diversity was observed in the horn of Africa (median heterozygosity = 0.70) and the highest in Oceania (median heterozygosity = 0.81) (Table 3, Fig. 4a). Similar trends occurred in effective cardinality (roughly equivalent to the number of alleles, with adjustment for minor allele frequency), with median values ranging from 3.88 in Africa to 5.18 in Oceania (Table 3, Fig. 4b).

Within-host diversity estimates using the high-diversity 100 microhaplotype panel were generated for each isolate using Complexity of Infection (COI) measures and correlated with the genome-wide estimates derived from the $F_{WS}$ score. Using thresholds of COI = 1 and $F_{WS} \geq 0.95$ to define a likely monoclonal infection, there were no significant differences in the proportion of polyclonal infections defined by the microhaplotype and genome-wide data in any geographic region (Supplementary Table 1). Furthermore, as illustrated in Fig. 5, the $F_{WS}$ scores of the infections with COI = 1 was significantly larger than those with COI > 1 in all regions, demonstrating a strong alignment between the microhaplotype and genomic data (Supplementary Table 1).

The capacity of the high-diversity microhaplotype panel to capture spatial transmission patterns including differentiation of geographically distinct populations was also assessed, using Principal Coordinate Analysis (PCoA) and IBD-based networks. The microhaplotype-based PCoA trends were consistent with spatial trends observed with genome-wide datasets[24]. Analysis of all the available *P. vivax* data demonstrated three major clusters with PC1 and PC2 representing South America, Africa and West Asia (group 1), East and West Southeast Asia (group 2), and Oceania and Maritime Southeast Asia (group 3) (Fig. 6a, b). Clear differentiation of the regional groups was observed within each of the major clusters. To retain accuracy in MAF estimates, IBD-based analyses were restricted to within each of the 7 regional groupings. In Maritime Southeast Asia, a major clonal outbreak in Malaysia (defined K2) that we have described previously with genomic data[30], was also distinguished clearly with the microhaplotypes (Fig. 6c, d). Furthermore, the resolution of previously defined K3 and K4 Malaysian sub-populations was also captured with the microhaplotypes. In other geographic regions, patterns of infection relatedness were largely consistent with genomic patterns described in previous studies (Supplementary Fig. 4)[24].

Further spatial evaluations of the high-diversity 100 microhaplotype panel were undertaken to assess the accuracy of such panels in detecting imported *P. vivax* cases by determining an infection's country of origin. Using the data from 21 countries with a minimum sample size of 4 from the available *P. vivax* global dataset, we determined the accuracy of the 494 SNPs within the microhaplotype panel in predicting country of origin using a recently developed Bi-Allele Likelihood (BALK) classifier[31]. The predictive accuracy of the 494 microhaplotype SNPs was compared to the 38-SNP Broad barcode and

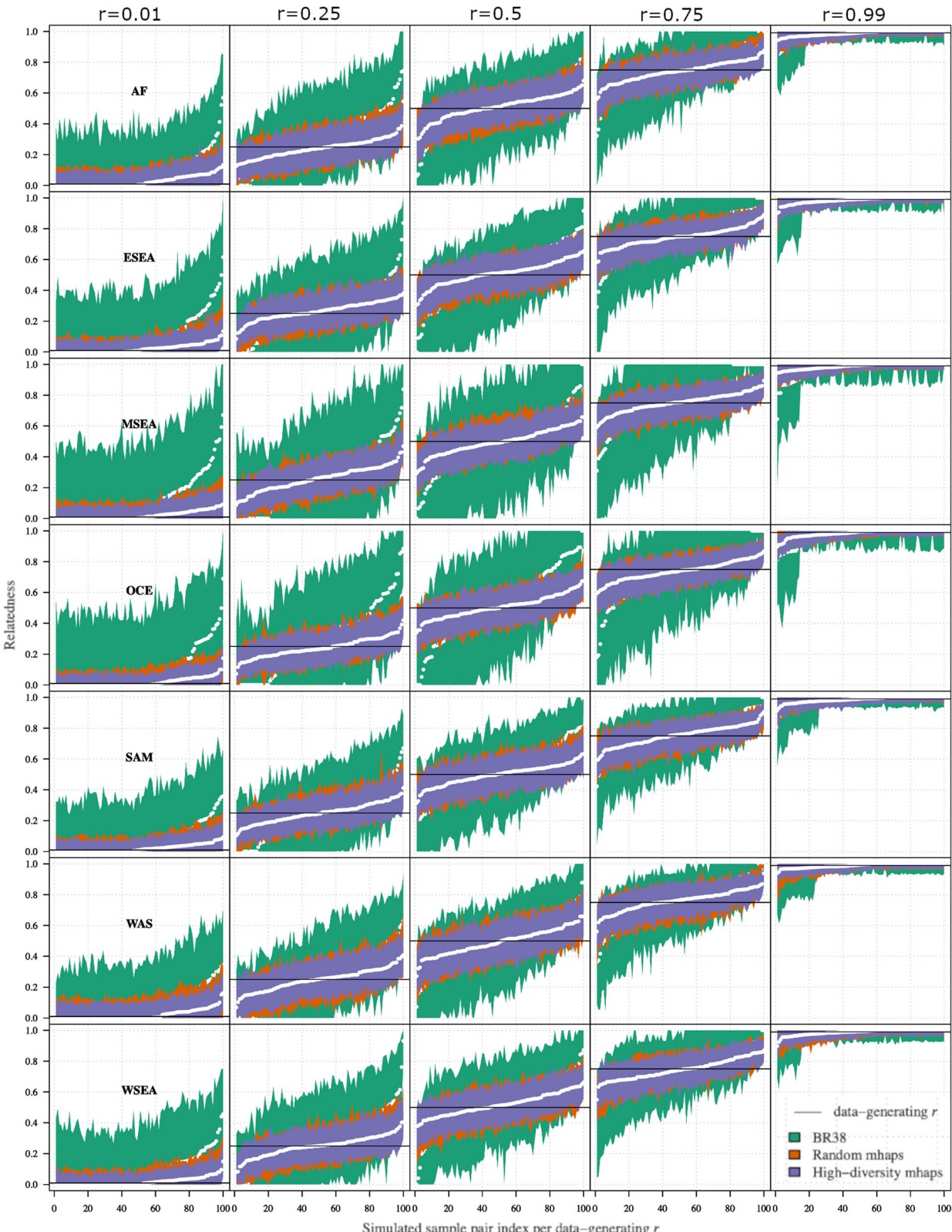

**Fig. 3 | Confidence intervals around infection relatedness estimates using 100 microhaplotype-based simulations.** The 95% confidence intervals (CI, shaded regions) surrounding the mean are based on data simulated using 5 data-generating relatedness parameters, *r*. Data are presented on 3 marker panels: High-diversity microhaplotype panel (purple), Random-SNP microhaplotype panel (orange), and 38 Broad barcode biallelic SNPs (green). Separate plots are provided for each *r* (from 0.01–0.99) and geographic region; AF (Africa), ESEA (East Southeast Asia), MSEA (Maritime Southeast Asia), OCE (Oceania), SAM (South America), WAS (West Asia) and WSEA (West Southeast Asia). The data simulations were conducted on allele frequency data input from $n = 34$ (AF) $n = 151$ (ESEA), $n = 59$ (MSEA), $n = 120$ (OCE), $n = 135$ (SAM), $n = 31$ (WAS), and $n = 85$ (WSEA) independent, monoclonal samples.

**Table 2 | Genome-wide versus microhaplotype-based identity by descent (IBD) estimates in *P. vivax* infection pairs**

| Initial infection | Recurrent infection | Patient sampling framework | Site | Region | Year | No. days until recurrence | Genome-wide IBD fraction (*recurrence classification) | Microhaplotype IBD fraction (*recurrence classification) |
|---|---|---|---|---|---|---|---|---|
| VVX01694 | VVX01695 | Clinical survey: TQ vs PQ[26] | Oddar Meanchey, Cambodia | ESEA | 2015 | 43 | 1.000 (clone) | 1.000 (clone) |
| VVX01743 | VVX01744 | Clinical survey: TQ vs PQ[26] | Oddar Meanchey, Cambodia | ESEA | 2015 | 169 | 1.000 (clone) | 1.000 (clone) |
| VVX01078 | VVX12129 | Clinical survey: 7- vs 14-day PQ[27] | Dak O, Vietnam | ESEA | 2015 | 332 | 0.997 (clone) | 0.973 (clone) |
| VVX01026 | VVX12145 | Clinical survey: 7- vs 14-day PQ[27] | Krong Pa, Vietnam | ESEA | 2015 | 130 | 0.070 (distant relative) | 0.110 (distant relative) |
| VVX12151 | VVX12078 | Clinical survey: 7- vs 14-day PQ[27] | Krong Pa, Vietnam | ESEA | 2016 | 53 | 1.000 (clone) | 1.000 (clone) |
| VVX12078 | VVX12150 | Clinical survey: 7- vs 14-day PQ[27] | Krong Pa, Vietnam | ESEA | 2016 | 49 | 1.000 (clone) | 1.000 (clone) |
| VVX01303 | VVX01304 | Clinical survey: TQ vs PQ[26] | Bangkok, Thailand | WSEA | 2014 | 61 | 0.999 (clone) | 0.977 (clone) |
| PD0607-C | PD0623-C | Clinical survey: AS vs CQ vs CQ+PQ[28] | Tak, Thailand | WSEA | 2001 | 136 | 1.000 (clone) | 0.977 (clone) |
| PY0023-C | PY0023-CW | Clinical survey: CQ vs AM[29] | Sabah, Malaysia | MSEA | 2014 | 12 | 1.000 (clone) | 0.975 (clone) |
| PY0109-CWx | PY0109-Cx | Clinical survey: CQ vs AM[29] | Sabah, Malaysia | MSEA | 2013 | 29 | 1.000 (clone) | 0.937 (clone) |
| PY0081-C | PY0067-C | Clinical survey: CQ vs AM[29] | Sabah, Malaysia | MSEA | 2014 | 29 | 1.000 (clone) | 1.000 (clone) |
| PY0082-C | PY0072-C | Clinical survey: CQ vs AM[29] | Sabah, Malaysia | MSEA | 2014 | 28 | 1.000 (clone) | 1.000 (clone) |
| PJ0005-C | PJ0009-C | Returning travellers** | Papua Indonesia | OCE | 2010 | 861 | 0.036 (stranger) | 0.000 (stranger) |
| PJ0167-C | PJ0166-C | Returning travellers** | Papua Indonesia | OCE | 2013 | 71 | 0.048 (stranger) | 0.156 (distant relative) |

List of sample pairs evaluated for relatedness as a fraction of the genome IBD using genome-wide and microhaplotype-based data. Microhaplotype-based analyses were conducted using the High-diversity 100 microhaplotype panel. IBD fractions were determined using *hmmIBD* software, with microhaplotype variants analysed as biallelic SNPs. Clinical survey treatments; TQ (tafenoquine), PQ (primaquine), AS (artesunate), CQ (chloroquine), AM (artesunate-mefloquine). *Recurrence classification: clone (IBD ≥ 0.95), close relative (0.25 ≤ IBD < 0.95), distant relative (0.05 ≤ IBD < 0.25), stranger (IBD < 0.05). **Returning travellers presenting at the Royal Darwin Hospital, Darwin, Australia.

**Table 3 | Regional patterns of population diversity using the high-diversity microhaplotype panel**

| Region | Median heterozygosity (min-max) | Median effective cardinality (min-max) |
|---|---|---|
| AF | 0.74 (0–0.95) | 3.88 (1–18.94) |
| ESEA | 0.77 (0.55–0.95) | 4.41 (2.22–21.28) |
| MSEA | 0.76 (0.39–0.96) | 4.16 (1.65–23.91) |
| OCE | 0.81 (0.53–0.96) | 5.18 (2.12–26.05) |
| SAM | 0.80 (0.55–0.96) | 5.07 (2.2–27.16) |
| WAS | 0.70 (0.12–0.91) | 3.36 (1.13–11.72) |
| WSEA | 0.79 (0.48–0.97) | 4.84 (1.92–36.9) |

Statistics were calculated on $n = 615$ high-quality biologically independent monoclonal samples from the MalariaGEN Pv4 dataset.

three recently identified *P. vivax* geographic barcodes selected specifically for determining country of origin (GEO33, GEO50 and GEO55 comprising 33, 50, and 55 SNPs respectively)[31]. The median Matthew's

correlation coefficient (MCC, ranging from −1 when prediction is 0% correct to 1 when prediction is 100% correct) of the 494 microhaplotype SNPs were greater than or equal to the 38-SNP Broad barcode, GEO33 and GEO50 in 21 (100%) countries (Supplementary Table 2, Fig. 7). The GEO55 panel exhibited higher MCCs than the microhaplotype panel in 3/21 (14%) countries (Cambodia, Indonesia and Vietnam) but lower values in 4/21 (19%) countries (Afghanistan, India, Myanmar and Thailand). The median MCCs of the microhaplotype panel exceeded 0.9 in all countries except Cambodia and Vietnam (19/21, 90%); the median MCCs in these countries were also <0.9 at the 38-SNP Broad barcode and the three GEO panels.

**External validation of IBD accuracy using an exemplar high-diversity 100 microhaplotype panel with an independent dataset**

Given that the microhaplotype panels were selected using the MalariaGEN Pv4 dataset, we considered the potential of bias in using Pv4 data for the evaluation. An independent *P. vivax* genomic dataset was therefore employed for additional external evaluation of IBD

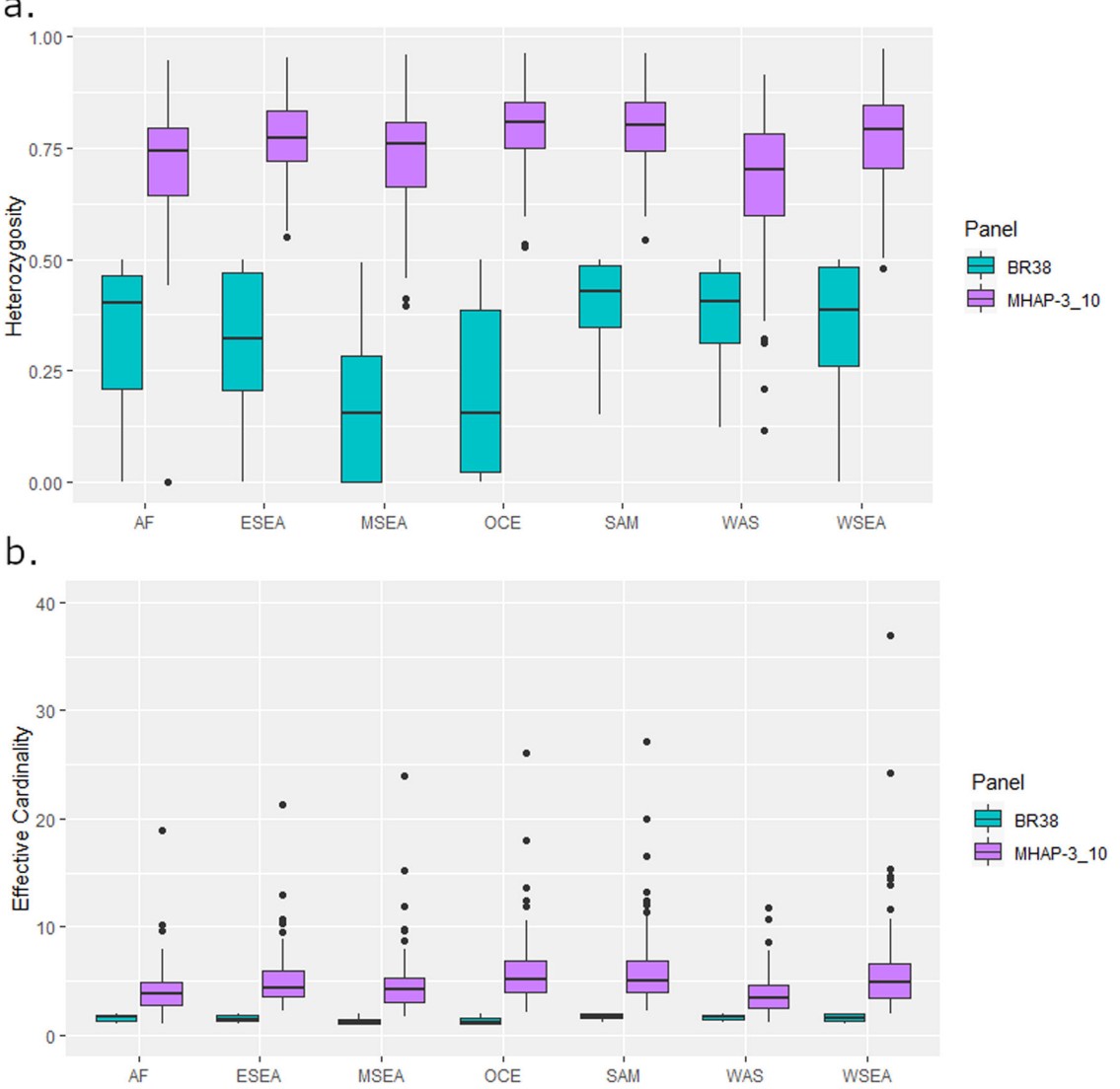

**Fig. 4 | Comparative diversity between the High-diversity microhaplotype panel and the 38-SNP Broad Barcode.** Panel (**a**) presents heterozygosity measures and panel (**b**) presents effective cardinality scores in $n = 615$ high-quality biologically independent monoclonal samples by panel and region. Each boxplot presents the median, interquartile range and min and max value for the hetereozygosity panel (**a**) and effective cardinality (**b**). Panel labels; 38-SNP Broad barcode (BR38) and High-diversity microhaplotype panel (MHAP-3_10). Regional labels; Africa (AF), East Southeast Asia (ESEA), Maritime Southeast Asia (MSEA), Oceania (OCE), South America (SAM), West Asia (WAS) and West Southeast Asia (WSEA).

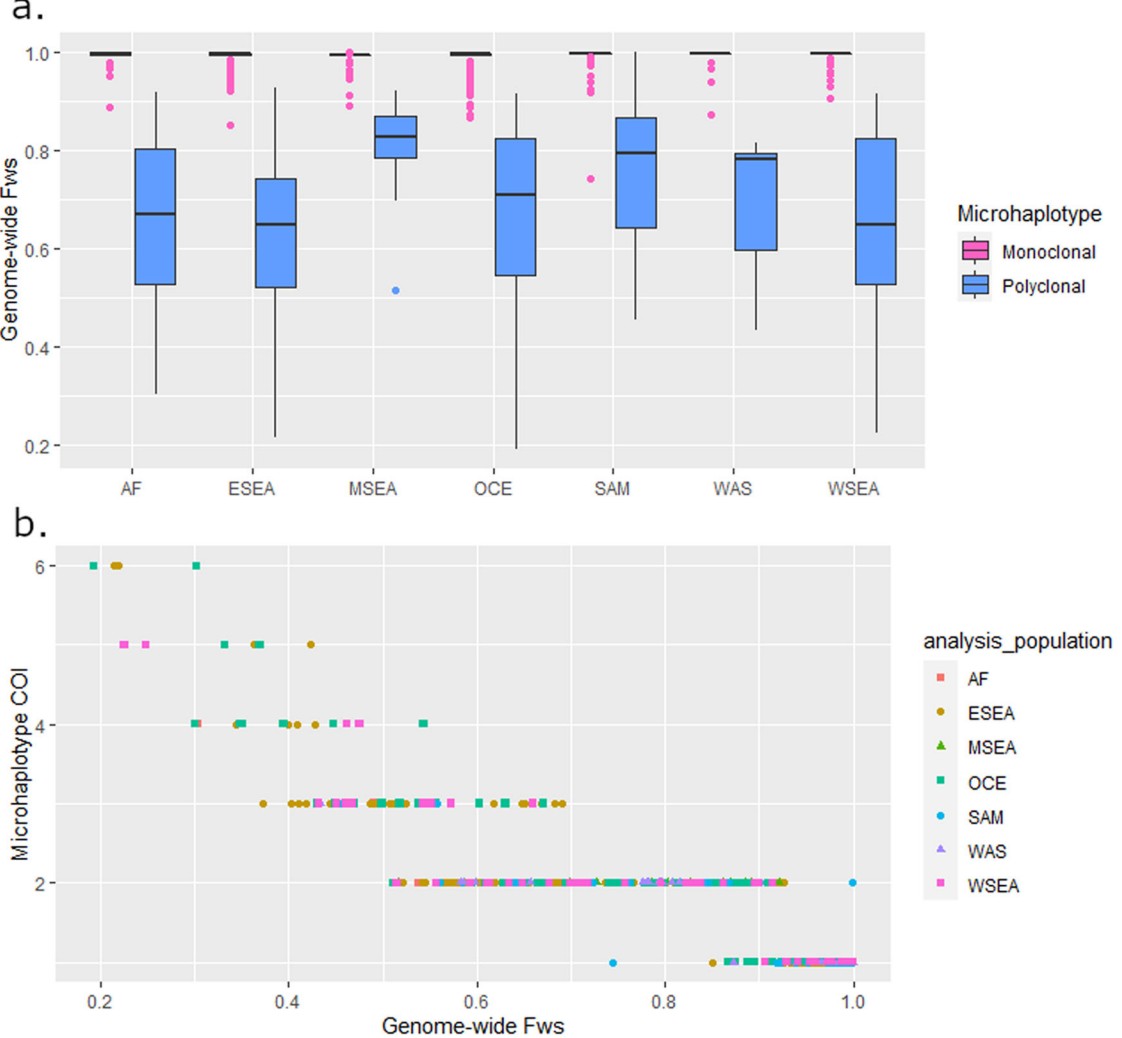

**Fig. 5 | Genome-wide $F_{WS}$ distribution by microhaplotype-based complexity of infection (COI).** Data from $n = 922$ high-quality biologically independent samples from Africa (AF), East Southeast Asia (ESEA), Maritime Southeast Asia (MSEA), Oceania (OCE), South America (SAM), West Asia (WAS) and West Southeast Asia (WSEA). Panel (**a**) provides boxplots illustrating the distribution of genome wide $F_{WS}$ scores in each of the monoclonal and polyclonal infection subsets as determined by THEREALMcCOIL analysis of the SNPs in the 100 microhaplotypes using the proportional function. Each boxplot presents the median, interquartile range and min and max of the $F_{WS}$. In all geographic regions, the median genome-wide $F_{WS}$ scores are closer to 1 (little to no within-host diversity) in the infections defined as monoclonal. Panel (**b**) illustrates the correlation between genome-wide $F_{WS}$ and microhaplotype-based COI estimates; a trend of decreasing COI is observed with increasing $F_{WS}$ (i.e., decreasing within-host diversity).

estimation using the high-diversity 100 microhaplotype panel. From an initial set of 836 non-Pv4 isolates, we identified 324 high quality, monoclonal samples[32–35]. The 324 samples derived from 19 endemic countries and clustered with infections from represented or geographically proximal countries in the Pv4 dataset (Supplementary Fig. 5, Supplementary Data 1[32–35]). Although all 7 geographic regions were represented amongst the 324 non-Pv4 samples, IBD analyses were restricted to 289 samples from 4 regional groups comprising ≥30 samples: AF ($n = 44$), ESEA ($n = 62$), SAM ($n = 95$) and WAS ($n = 87$). Pairwise measures of IBD were determined for the genome-wide SNPs and for the microhaplotype SNPs using *hmmIBD*. The 324 independent samples exhibited an overall higher proportion of missing calls at the 494 SNPs comprising the microhaplotype loci (median 17% SNPs, range 0–50% SNPs) than in the 615 Pv4 samples (median 0%, mean 2.5%, range 0–34%) reflecting the use of Pv4 for the original marker selection. Nonetheless, the microhaplotype-based estimates of pairwise IBD in the independent dataset demonstrated a significant positive correlation with the genome-wide estimates in all 4 geographic regions (all $P < 0.05$, Spearman's rho statistic using a paired test, Supplementary Fig. 6).

### Evaluation of microhaplotype number on IBD estimation

A formal evaluation of the impact of panel size on the accuracy of estimating IBD was undertaken using a set of 50, 100, 150, 200, and 250 microhaplotype panels generated using the greedy algorithm (see Notebook 2). Analysis was undertaken on monoclonal *P. vivax* infections using simulated data generated on pairs of infections across a range of relatedness ($r$) values using *paneljudge*[36]. Trends in marker panel performance were observed using the root mean squared error (RMSE) of the estimates of $r(\hat{r})$ compared to the data-generating $r$ (Supplementary Fig. 7), which shows that increasing panel size improves the accuracy of IBD estimation, with a large gain in panel informativeness between 50 and 100 microhaplotypes but diminishing returns above 100 microhaplotypes.

### Discussion

Our study provides the first description in silico of *P. vivax* microhaplotype panels which can be used to estimate identity by descent relatedness between pairs of acute and recurrent infection isolates, and thus help to discriminate between different causes of vivax malaria recurrence. A systematic genome-wide selection process was used to

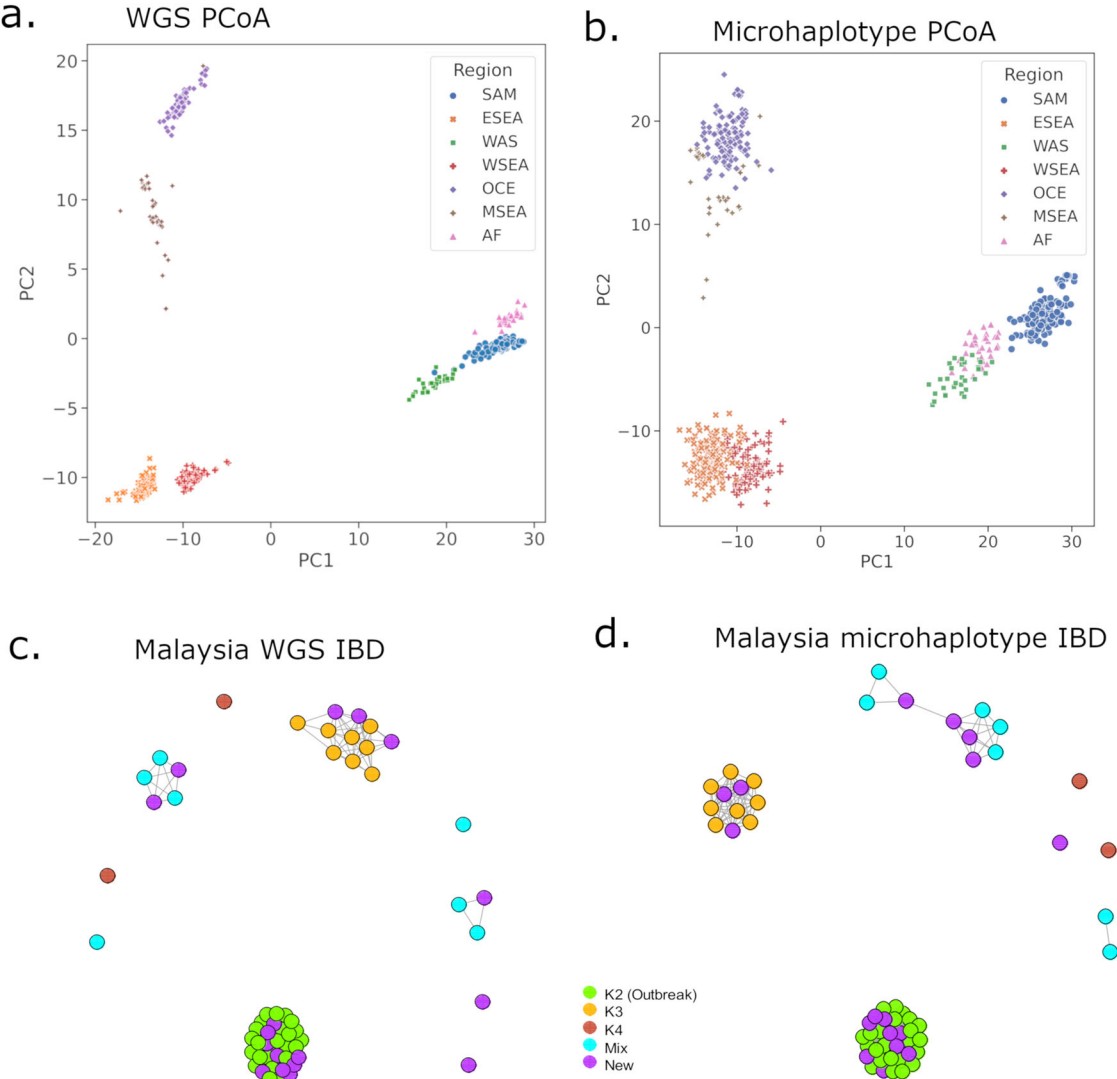

**Fig. 6 | Spatial trends in *P. vivax* connectivity using microhaplotypes versus genomic data.** Panels (**a**) and (**b**) present PCoA plots generated with whole genome sequencing (WGS) and High-diversity SNP microhaplotype data on *n* = 615 high-quality biologically independent monoclonal isolates from Pv4. The WGS data was generated at 574,604 biallelic SNPs (with minimum allele count ≥2), with PC1 (49.9%) and PC2 (17.5%) displayed. The microhaplotype plot also presents PC1 (46.4%) and PC2 (25.3%). The combinations of PC1 and PC2 provide marked separation of all 7 regional groups in both the WGS and microhaplotype data sets. Panels (**c**) and (**d**) present WGS and microhaplotype-based IBD infection networks in Malaysia. The network plots were generated at a set of 224,612 SNPs (WGS) and the High-diversity SNP microhaplotype panel in single clone Malaysian infections (*n* = 57) at a connectivity threshold of minimum IBD 0.5 (siblings or greater relatedness). Infections are colour-coded according to sub-structure definitions based on previously described ADMIXTURE analysis with genomic data[30]. Infections defined as "New" (purple) were not available in the previous analysis. The clustering patterns of the WGS and microhaplotype-based data are highly consistent; both data sets capture high connectivity amongst the K2 outbreak strains, a distinct K3 sub-population, and divergent K4 infections[27]. The new infections appear to derive from across the different sub-populations.

identify panels of 50–250 globally diverse microhaplotypes using an expansive *P. vivax* genome dataset. The utility of these panels was assessed using both simulated and 'real' genomic data, including an independent validation dataset. Panels of 100 or more microhaplotypes demonstrate significant potential to improve the interpretation of clinical trials and surveillance data. Further details on the integration of microhaplotype data in the proposed use cases, and areas requiring further research and development are described herein.

A key requirement of the *P. vivax* microhaplotype panels was the ability to estimate IBD accurately between paired isolates and thereby help to classify the likely origin of vivax malaria recurrence (i.e., relapse, recrudescence, or reinfection) in clinical trials. Our simulations showed that both high-diversity and random microhaplotype panels yielded consistently higher accuracy in IBD estimation (across

different IBD levels and in different populations) than the 42-SNP Broad barcode that is currently the most widely used SNP barcode for *P. vivax*. This is not surprising as the microhaplotypes have substantially greater genetic information content compared with the 38 evaluable BR38SNP panel. We also demonstrated greater efficacy of IBD estimation when microhaplotypes were selected to meet specified diversity criteria (high-diversity SNP panel) relative to microhaplotypes with random SNP selections (any-SNP panel). In addition, we were able to confirm that panel size increases informativeness greatly between 50 and 100 microhaplotypes per panel, with diminishing returns beyond 100. The marginal gains to informativeness at higher panel sizes (150, 200, 250) are consistent with modelling studies by Taylor and colleagues and come at a trade-off in cost and laboratory practicalities. Panels of 100 microhaplotypes provide a reasonable compromise.

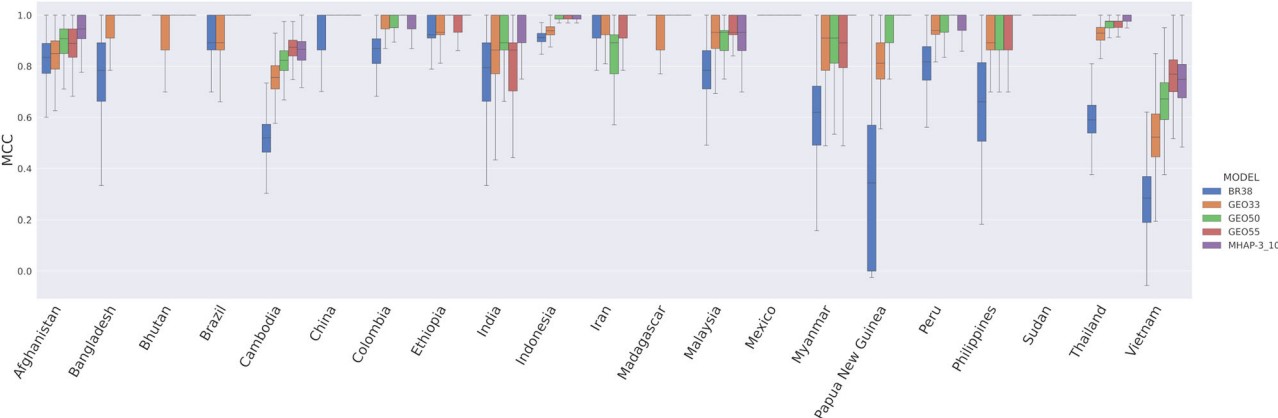

**Fig. 7 | Comparison of country prediction performance between SNP panels.** Comparisons were undertaken between the 494 SNPs in the High-diversity microhaplotype panel (MHAP-3_10), the 38-SNP Broad barcode (BR38), and the 33-, 50- and 55-SNP GEO panels (GEO33, GEO50 and GEO55 respectively). The boxplots present the Mathews correlation coefficient (MCC) scores from 500 repeats with stratified 10-fold cross validation for each SNP set using the Bi-Allele Likelihood classifier[31]. Each boxplot provides the median, interquartile range and min and max MCC for the given country and panel. Country labels are provided on the y-axis. Each bar presents the median, interquartile range and minimum and maximum MCC for the given country and model. The analyses were based on $n = 799$ biologically independent samples from 21 countries (each with $n \geq 4$ samples).

In accordance with previous predictions[21], our simulation-based results demonstrated error rates (RMSE) below 0.1 in the estimation of pairwise IBD in all populations tested using the high-diversity microhaplotype panel. The error rates were highest for the prediction of infections with 50% IBD, which is consistent with previous findings. Further work is needed to understand how error rates might impact clinical predictions of individual treatment response or population-level drug efficacy both in the context of descriptive data analyses and mathematical models[37]. In descriptive analyses, data summaries (e.g., allelic homology observations or relatedness estimates) and a set of rules (e.g., relatedness values greater than 25% are suggestive of relapse) are used to classify recurrences categorically. Rules-based classification is not the same as estimating the probability of relapse given the data under a statistical model (as in Taylor et al.[21], which also accounts for the added complexity associated with multiclonal infections). Nevertheless, improved data informativeness for IBD estimation likely translates into improved model-based probabilistic classification performance, at least in the case of the model of Taylor, because that model machinery includes an intermediary that evaluates the probability of the data given IBD partitions compatible with networks of sibling, clonal and stranger parasites[21].

It should be acknowledged that panel evaluation based on simulated data was done in a highly idealised manner that captures panel performance in its most favourable light. Data were simulated under the same model used to estimate relatedness, such that the model was perfectly specified in relation to the simulated data. Real data are generated by an ancestral process that the model does not capture, i.e., it is mis-specified in relation to real data. The impact of mis-specified allele frequencies, mis-classified multiclonal infections, etc. was not evaluated. In addition, it should be acknowledged that genotyping errors and failures (missing data) were not included in our simulations as this would require reprogramming of the *paneljudge* package, which is beyond the remit of the current study. However, we anticipate that genotyping failures should be infrequent with NGS-based amplicon sequencing methods, where each locus is typically covered by hundreds of reads[23].

Data from pairs of *P. vivax* isolates, collected from recurrent infections in clinical trials, confirmed the ability of the high-diversity microhaplotype panel to estimate pairwise IBD. High correlations were observed between microhaplotype and genome wide IBD estimates in a set of 14 paired *P. vivax* isolates from the same patient before and after treatment. Using our assigned IBD thresholds, only one of 14 pairs of infections had mismatching classifications of stranger versus distant relative, which are both likely to reflect either reinfections or relapses. However, the available genomic data did not comprise any infection pairs with IBD estimates around 50% (i.e., siblings), which our simulations predicted to be the most difficult relationships to determine accurately. The number of genomic pairs was limited owing to the difficulty in obtaining enough DNA and thus high-quality sequence data from recurrent infections, which typically exhibit low parasite densities. This is a major incentive for using targeted genotyping approaches. Nonetheless, we demonstrated high correlations between the microhaplotype and genome-wide IBD estimates in the assessments of day 0 samples, where a range of IBD relationships were observed.

In some scenarios, genetic data alone will not be informative of the likely origin of a recurrence. A single mosquito inoculation may carry clonal parasites or related parasites generated through the recombination of heterologous clones. If a human host carries hypnozoites from a single inoculation, the relapsing parasites will therefore either be homologous or related (meiotic siblings) to the incident infection. In situations where the relapses are homologous, they will be genetically indistinguishable from recrudescence. If a host carries hypnozoites from one or more previous mosquito inoculations, relapsing parasites are liable to be unrelated/heterologous to the incident infection, and thus genetically indistinguishable from reinfections. In our study, a high proportion of infection pairs (79%) had homologous genomes, although the majority (86%) of pairs derived from low seasonal transmission settings in Malaysian Borneo, southern Vietnam, Cambodia and western Thailand. A therapeutic efficacy study undertaken in Peru also identified a moderately high proportion of genetically homologous recurrences (52%, 12/23)[12]. The proportion of relapses amongst the recurrences in our and the Peruvian study is uncertain. In a therapeutic efficacy study undertaken in Cambodia, 20 patients were relocated to a malaria-free area excluding the possibility of reinfection; the authors were able to confidently define five recurrences as relapses and demonstrated that 4 (80%) of these were related to the initial infection[38]. As more comprehensive data become available, a clearer picture of the epidemiology of recurrent *P. vivax* infections will emerge. Even in areas where the prevalence of related recurrences is low, mathematical modelling approaches that combine genetic data with time-to-event data will help to resolve the probable cause of recurrence[37].

A recent study highlights the human spleen as an important reservoir of *P. vivax* parasites[39]. However, there is currently no evidence of altered metabolism of endosplenic stages in human *P. vivax* infections, in marked contrast to the hypnozoite reservoir. We therefore anticipate that the endosplenic stages are treatable by blood-stage antimalarials such as chloroquine, although this needs to be confirmed. In this context, information on IBD between paired clinical isolates would not distinguish between endosplenic and circulating infections but should still help to distinguish reactivated liver stages and reinfections.

Another key requirement of the microhaplotype panel was the ability to capture spatial *P. vivax* transmission dynamics. Strategies to contain *P. vivax* effectively will be assisted by a more comprehensive understanding of the major routes of infection spread within and across borders. The SNP-based data from our microhaplotype panel displayed clear geographic trends and high accuracy in predicting the country of origin, suggesting utility in detecting and mapping imported *P. vivax* cases. Although the high-diversity microhaplotype panel was not intentionally selected for country prediction, the rich genetic data enabled high-performance country prediction compared to recently described geographic marker panels (GEO33, GEO50, and GEO55)[31]. The equivalently high prediction accuracy of the micro-haplotypes relative to the GEO panels means that users who want to capture information on both IBD and on country of origin can derive all information with the microhaplotypes alone without impeding on accuracy. The accuracy in pairwise IBD estimation using the micro-haplotype data also demonstrates a unique potential for tracking infection spread at micro-epidemiological spatial resolution, to inform the dispersal of infections within and between communities. For example, the microhaplotype data from Malaysia effectively captured a previously described clonal expansion, as well as more subtle population structure reflecting different foci of infection. The spatial analyses conducted here used biallelic SNP data at the micro-haplotypes; whilst the high density of SNPs (n = 494) provided rich genetic information, even greater information content can be achieved using multiallelic microhaplotypes once new software to deal with these complex datasets becomes available. It is possible that some microhaplotypes are subject to site-specific selective pressure, which could affect their spatio-temporal surveillance utility; this impact could be mitigated by implementing additional microhaplotypes.

Some global regions are not well represented in the Pv4 genomic dataset that were used for marker selection, such as Africa, the Indian subcontinent (West Asia), Central and South America (SAM). It is therefore unclear how well the microhaplotypes described here will capture IBD in these regions. Some insights have been derived from our evaluation of an independent (non-Pv4) *P. vivax* genomic dataset. Amongst the four regional groups evaluated in the independent dataset moderate differences were observed in country representation. For example, the Pv4 Africa dataset only comprised Ethiopian isolates, whilst the independent dataset comprised six African countries. There were also differences between the two datasets in four countries in South America, three in East Southeast Asia and two in West Asia. Despite the differences in country representation, the independent data regional groups exhibited significant correlations between genome-wide and microhaplotype-based estimations of IBD. This suggests that the panel discovery framework captured globally representative microhaplotypes from the Pv4 dataset. Nonetheless, the informatics framework that was established for the micro-haplotype selection can be applied readily to update panels as needed once additional whole genome data become available from new geographical regions. The framework can also be used to select country- or region-specific panels where needed.

Information on within-host infection complexity is important to capture epidemiologically relevant transmission dynamics. We observed high concordance in the proportion of polyclonal infections captured by microhaplotype-based COI and genome-wide $F_{WS}$ measures when thresholds of COI = 1 and $F_{WS} \geq 0.95$ were applied. Only a few infections displayed differences in the classification of poly-clonality between the microhaplotype and genomic datasets. In interpreting these differences, it should be acknowledged that the 0.95 $F_{WS}$ threshold is only a guideline, and that population distributions of within-host diversity generally reflect a continuum, not discrete clusters of monoclonal and polyclonal infections.

Within-host diversity at SNP barcodes also follows a continuum, but this is less marked owing to the lower genetic resolution. When monoclonal thresholds were only applied to the COI data, the $F_{WS}$ demonstrated significantly larger values in the COI = 1 (monoclonal) vs COI > 1 (polyclonal) infection group in all geographic regions investigated, highlighting consistency between the two data sets. Further work is needed to determine how to phase microhaplotype profiles in highly complex infections where any number of clones may be present in varying proportions, but tools such as the *Dcifer* software provide an important step forward[40]. Indeed, phasing of polyclonal malaria infections is not a challenge unique to microhaplotypes.

The *P. vivax* genome has an abundance of globally diverse microhaplotype regions that can effectively capture information on infection lineages and spatial connectivity, overcoming the previous requirement to generate genomic data in samples that are often notoriously difficult to sequence. With targeted, deep sequencing platforms, these markers have great potential to inform on the complex diversity within individual infections and associated insights on transmission. The establishment of targeted microhaplotype geno-typing tools for *P. vivax* will transform the assessment of clinical surveys in this species, enhance knowledge of relapse biology, and greatly improve surveillance of infections.

## Methods

### MalariaGEN Pv4 Data preparation

The initial dataset comprised 1,816 samples from 17 countries derived from the *P. vivax* community study (Pv4) in the vivax and Malaria Genomic Epidemiology Networks (vivaxGEN and MalariaGEN)[24], as well as previously published external studies[41,42]. Of the 1,895 samples described in Pv4 data resource, the 1,816 samples reflect data for which we had access prior to the open release. All genomic datasets were generated using Illumina short-read sequencing platforms. Sequence alignment, SNP discovery and variant calling, population assignments, and $F_{WS}$ (within-sample F statistic) calculations for within-host allele infection complexity were undertaken using previously described methods within the MalariaGEN framework, producing a dataset referred to as *P. vivax* Genome Variation Project release 4.0 (Pv4). As described in the data resource, the Pv4 data set comprises ~4.5 million variants of which 911,901 are high-quality biallelic SNPs suitable for population genetic analyses[24]. The $F_{WS}$ estimates the fixation of alleles within each infection relative to the diversity observed in the total population on a scale from 0 to 1 and were provided as part of the Pv4 dataset[24,43]. Using a threshold of $F_{WS} \geq 0.95$ as a proxy to identify a monoclonal infection, all polyclonal infections were excluded from subsequent analysis. High-quality samples were selected using a threshold of ≥50% core genome, notated by the "analysis-set" flag in the MalariaGEN Pv4 dataset. The patient metadata provided by the contributing VivaxGEN partner studies was used to identify independent isolates for marker selection, and recurrent isolates for downstream evaluation of selected marker sites. The MalariaGEN curated metadata was used to define 7 regional-level geographic assignments with the following categories: Africa (AF), South and Central America (SAM), West Southeast Asia (WSEA), East Southeast Asia (ESEA), Maritime Southeast Asia (MSEA), Oceania (OCE) and West Asia (WAS). Rather than using country-based classifications to define infection groups, we employed the regional classifications defined in the MalariaGEN Pv4 data resource, which are based on both geographic

location and genomic clustering patterns. This approach is more accurate than purely country-based classifications since malaria transmission networks do not always conform to national spatial boundaries. For example, there is a malaria-free 'corridor' that runs through the middle of Thailand, leading to distinct separation of eastern and western infections[44]. In contrast, some border regions, such as parts of Vietnam and Cambodia, are very 'porous' with high levels of homology seen between infections across national borders.

**External Validation data preparation**
A genomic dataset with non-overlapping samples with the MalariaGEN Pv4 dataset was established for independent validation of IBD estimation accuracy in microhaplotypes selected from the Pv4 dataset. Details on the Methods can be found in Supplementary Note 1. In brief, the independent dataset was sourced from all published, open-access, Illumina paired-end genome-wide *P. vivax* data that was not represented in Pv4. A total of 836 independent samples were sourced from 8 studies[32–35,38,45–47]. Read alignment, variant calling and genotype calling were undertaken in close alignment with the MalariaGEN Pv4 pipeline with the aim of deriving comparable genotype calls at the 911,901 high-quality biallelic SNPs detected in Pv4[24]. Samples were filtered to remove those with >50% missing variants, and variants were filtered to remove those with >25% missing samples. The resulting variant calling format (vcf) file comprised 329,371 variants in 324 samples. The R-based *moimix* package was used to calculate the $F_{WS}$ and a threshold of ≥0.95 was used to assign monoclonal infections for downstream clustering and IBD analyses[48]. The high-quality monoclonal Pv4 dataset was subset to the 329,371 variants. A genetic distance matrix was constructed on the pooled Pv4 and non-Pv4 genome-wide dataset using the neighbour-joining method implemented with the R-based *ape* package[49]. The neighbour-joining tree was visually inspected to define regional classifications for the independent dataset samples.

**Marker selection**
We tailored our marker selection (Fig. 1) to the Illumina amplicon-based sequencing workflow as this methodology is widely used in the malaria community and has demonstrated utility and feasibility for malaria molecular surveillance in low- and middle-income countries[17]. The maximum amplicon size for Illumina amplicon-based sequencing is 200 bp, which set the criteria for the maximum length of each microhaplotype. The minimum target of 100 microhaplotypes was based on a mathematical modelling study, which demonstrated that under idealised settings (i.e., estimating relatedness from data simulated under the model used for estimation), IBD estimates with low root mean squared error (RMSE < 0.1) between monoclonal malaria parasites with relatedness equal to 0.5 (relatedness with the highest RMSE) are obtainable using ~100 polyallelic markers[21]. Diminishing returns in RMSE reduction were observed above 100 polyallelic markers, highlighting this target number as a pragmatic balance of IBD accuracy against the economic cost of primers and assays[21]. Panels of microhaplotypes (schematic Fig. 1b) were subsequently selected to optimise marker selection with the previously identified criteria using a three-phase approach as described in Fig. 1a with respect to sample, variant and window selection. In the first phase, samples were subset from the MalariaGEN Pv4 data resource that consisted of high-quality samples as identified in Pv4 as being independent, 50% of the genome callable, and likely monoclonal with Fws ≥ 0.95. In the second phase, candidate SNPs were identified by filtering the high-quality biallelic SNPs (filter pass), with minor allele frequency (MAF) 0.1 and genotype failure rates <0.1. Finally, candidate microhaplotype windows were identified by scanning the core genome (excluding hypervariable regions) in coding regions with 50 bp overlapping, 200 bp windows to identify amplicon-sized segments containing one or more candidate SNPs. We then defined the Pv heterozygome as the collection of all

windows that additionally have sufficiently high diversity (windows with a minimum heterozygosity of 0.5 and at least 3 SNPs (3,830 total windows). A Manhattan plot of all windows distributed across the core, coding regions of the *P. vivax* genome is shown in Fig. 1c. Microhaplotype panels were then selected from the *P. vivax* heterozygome by further selecting windows that had between 3–10 SNPs, and approximately evenly distanced spacing across 14 chromosomes and a minimum heterozygosity of 0.6 (1,110 windows total). The Random mhaps panel had 100 markers chosen irrespectively of diversity and the high-diversity 100 SNP panel had each marker chosen with the highest possible heterozygosity in a given genomic region (Fig. 2c). Five additional panels were generated to separately evaluate the impact of marker number (50, 100, 150, 200 and 250 marker panels) which were selected with the same criteria as the high-diversity panel using the greedy algorithm described in notebook 2: https://github.com/svsiegel/vivax-mhaps. Measures of MAF and heterozygosity were calculated using the scikit-allel package (microhaplotype discovery marker framework, notebook 1: https://github.com/svsiegel/vivax-mhaps).

**Population genetic evaluation of a 100-microhaplotype panel**
A panel of 100 high-heterozygosity microhaplotypes (high-diversity panel) was selected as an exemplar microhaplotype panel for comparative evaluation against the Broad *P. vivax* barcode, as well as a panel that had similar characteristics without being specifically optimised for highest possible heterozygosity (random panel). These two panels were discovered from a broader set of microhaplotypes output from the greedy algorithm and a final selection was curated manually. (Supplementary Data 2). Population diversity was assessed at each panel (microhaplotype and Broad) for each of the 7 geographic regions using measures of effective cardinality and heterozygosity computed with the R-based *paneljudge* package (release not published) available at https://github.com/aimeertaylor/paneljudge. Analyses were conducted on the high-quality monoclonal samples using one of the two alleles selected at random to reconstruct microhaplotypes at heterozygote genotype positions using the haploidify function in Scikit-allel version 1.2.0 (https://github.com/cggh/scikit-allel). Measures of within-host diversity were also generated using *THE REAL McCOIL* package version 2.0 (https://github.com/EPPIcenter/THEREALMcCOIL) on the biallelic SNPs in the High-diversity microhaplotype panel and correlated against the genome-wide $F_{WS}$ estimates of infection complexity[43,50].

The potential of the biallelic SNPs within the microhaplotype panel in capturing spatial patterns of transmission was assessed using Principal Coordinate Analysis (PCoA) to illustrate the regional-level geographic clustering patterns. PCoA analysis was performed on the same high-quality monoclonal samples (n = 615) by conducting PCA measures using *sklearn.decomposition PCA* method of scikit-learn package version 1.2.0 on a microhaplotype-based pairwise distance matrix calculated from the proportion of non-identical microhaplotype alleles across all positions. Spatial clustering patterns were also assessed using IBD-based measures of differentiation implemented with the *hmmIBD* package as described below and plotted using the R-based *igraph* package version 1.6 (https://igraph.org).

A Bi-allele Likelihood (BALK) classifier was used to predict the country of origin of *P. vivax* isolates using SNP data (not multi-allelic microhaplotypes) at biallelic SNPs in the High-diversity SNP 100 microhaplotype panel, Broad *P. vivax* barcode, and three recently identified *P. vivax* geographic barcodes (GEO33, GEO50 and GEO55)[25,31]. Details of the BALK classifier can be found in the original paper[31]. Country prediction performance was assessed using data from 799 of the 922 high-quality samples in the available Pv4 genomic dataset. The 799 sample-set corresponds with the sample set used in the original paper describing the BALK classifier[31]. Briefly, the sample set was derived by filtering samples to include a single representative

of samples with near-identical genomes (defined as pairs with genetic distance <0.001), subjecting to iterative data quality filtering to obtain the best representative number of samples and informative SNPs without any missing genotype by removing samples with higher missingness iteratively, and then removing samples that appeared to be imported based on genome-wide data clustering patterns. A total of 21 countries had ≥4 samples. The comparative predictive performance of the SNP panels was evaluated using a stratified 10-fold cross-validation with 500 repeats, reporting the Matthews Correlation Coefficient (MCC).

### Evaluation of IBD estimation with using simulated data

The 2 selected 100 microhaplotype panels (random microhaplotype panel and high-diversity microhaplotype panel), as well as the 42-SNP Broad barcode panel, were evaluated with the *paneljudge* package for their ability to estimate relatedness (IBD) using a range of simulations to compute the uncertainty (error rate) in estimations of relatedness (*r*) ranging from 0 (strangers) to 1 (identical) under varying allele frequency distributions in different geographic regions. Details of the equations, assumptions, and parameter options in the *paneljudge* package can be found on GitHub (https://github.com/aimeertaylor/paneljudge). Briefly, for each of a range of data-generating *r* values (*r* = 0, 0.25, 0.5, 0.75 and 1.0), data were simulated on a pair of haploid genotypes, generating 100 haploid genotype pairs. Estimates of $r(\hat{r})$, switch-rate parameter $k(\hat{k})$, the 95% confidence intervals (CIs) around the estimates $\hat{r}$ and $\hat{k}$, and root mean square errors (RMSE) were then computed. The switch-rate parameter modifies the rate of switching between latent IBD and non-IBD states in the hidden Markov model by multiplying the probability of a crossover per base pair (default value 7.4e-7 as per[51]) and the inter-marker distance in base pairs. The accuracy of a given marker panel in estimating *r* was measured by the CI width and RMSE, ranging from ~0 (near perfect informativeness) to 1 (uninformative). The CI width and RMSE of $\hat{r}$ compared to the data-generating *r* never reach 0 as a genome of finite length may have many realised relatednesses compatible with a given probability of IBD. This means that there will always be some variance in realised relatedness around *r*. For comparative value, *paneljudge* simulations were run on two panels of 100 microhaplotype markers, one optimised for heterozygosity (high-diversity panel), one not optimised for heterozygosity (random panel), and for a set of 38 assayable biallelic SNPs from the 42-SNP Broad barcode (BR38) that has been widely used by the *P. vivax* surveillance community[25,52,53]. The five additional "greedy" algorithm panels consisting of 50, 100, 150, 200, and 250 markers were also analysed for RMSE comparisons against panel size using the same method (Supplementary Fig. 3).

### Evaluation of IBD estimation using real data

In addition to simulation-based data, the efficacy of the high-diversity 100 microhaplotype panel in estimating IBD was assessed relative to gold-standard genome-wide data using high-quality genomic data (≥50% genome covered by at least 5 reads) from monoclonal infections ($F_{WS} \geq 0.95$). Evaluations were undertaken in the MalariaGEN Pv4 data resource using both baseline (day 0 infections) data, and pairs of day 0 and recurrent infections, and in the independent validation data set using baseline data. IBD estimates between infection pairs were generated using the *hmmIBD* version 3.0 software[54]. As the version 3 software does not enable adjustments in the genotyping error rate for polyallelic markers (which may reduce artificially the IBD estimates between infections) the microhaplotype data were run as biallelic SNP markers rather than polyallelic markers. Sample pairs were run in geographically defined batches using allele frequency estimates based on the regional estimates from the baseline samples (high-quality, monoclonal, independent infection) using default parameters. Genome-wide and microhaplotype-based recurrence classifications were assigned using the following infection pair IBD thresholds; clone (pairwise IBD ≥ 0.95), close relative (0.25 ≤ IBD < 0.95), distant relative (0.05 ≤ IBD < 0.25), stranger (IBD < 0.05). The concordance between the microhaplotype and genome-wide IBD estimates was evaluated by the correlation coefficient between the datasets, and the proportion of recurrence classification mismatches.

### Reporting summary

Further information on research design is available in the Nature Portfolio Reporting Summary linked to this article.

## Data availability

Sequencing data from the MalariaGEN Pv4 samples has been made publicly available in the European Nucleotide Archive (ENA), with details provided in a data resource describing the MalariaGEN Pv4 data[24]. Accession codes are also available in Supplementary Data 1 and the VCF can be downloaded here: https://www.malariagen.net/resource/30. The sequencing data from the independent validation samples has also been made publicly available in the ENA or National Institutes of Health Sequence Repository Archive in the respective contributing studies[32–35]. Accession codes are also available in Supplementary Data 1.

## Code availability

The full microhaplotype marker discovery framework, which includes the development of an easy-to-use code and additional exploratory and optimisation analyses that are fully customisable can be found here: https://github.com/svsiegel/vivax-mhaps and the Github pages link https://svsiegel.github.io/vivax-mhaps/ with citable code for vivax-mhaps v1.0.0 using DOI:10.5281/zenodo.12622789. This repository can be easily accessed and run without needing to download the repository files, as it is connected directly to the Pv4 data package in a cloud-based instance using Google Colab, including all of the needed supplementary analysis files here https://svsiegel.github.io/vivax-mhaps/. This framework could additionally be applied to other data resources and other malaria species with minimal changes to the underlying codebase.

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

## Acknowledgements

The study was supported by the National Health and Medical Research Council of Australia (APP2001083 supporting S.A. and S.V.S.), the Wellcome Trust (200909 and ICRG GR071614MA Senior Fellowships in Clinical Science to R.N.P., 206194/Z17/Z supporting J.C.R. and S.V.S.) the National Institutes of Health (R01AI137154 to J.C.R.) and the Bill & Melinda Gates Foundation (INV-043618 supporting S.A. and R.N.P.).The whole genome sequencing component of the study was supported by the Medical Research Council and UK Department for International Development (award number M006212 to DK) and the Wellcome Trust (award numbers 206194 and 204911 to D.K.). The IMPROV clinical trial was supported by the Bill & Melinda Gates Foundation (OPP1054404 awarded to R.N.P.). We thank the patients who contributed their samples to the study, and the health workers and field teams who assisted with the sample collections. Genome sequencing was undertaken by the Wellcome Sanger Institute, and we thank the staff of the Wellcome Sanger Institute Sample Logistics, Sequencing, and Informatics facilities for their contribution.

## Author contributions

S.A., S.V.S., R.N.P., and J.C.R. conceived the study. S.V.S, H.T., R.A., and S.A. designed major components of the study. S.A., S.V.S, H.T., and R.A. wrote the original drafts for major sections of the manuscript. S.V.S, H.T., R.A., K.M., E.S., M.K, G.W., and S.A. conducted data analysis. A.R.T., J.A.W., D.P.K., J.C.R., and R.N.P. contributed critical guidance and tools for the analytical methods and data interpretation. D.P.K. and J.C.R. contributed sequencing, data production and informatic support. M.I., A.A., A.G.R, N.H.C., T.T.H., J.A.G., G.C.K.W.K., N.J.W., N.D., D.P.K., J.C.R., and R.N.P. contributed essential field-based malaria collections and metadata.

## Competing interests

The authors declare no competing interests.

## Additional information

Sasha V. Siegel [1,2], Hidayat Trimarsanto [2,3], Roberto Amato [1], Kathryn Murie[1], Aimee R. Taylor[4], Edwin Sutanto [5], Mariana Kleinecke [2], Georgia Whitton[1], James A. Watson [6,7], Mallika Imwong [8], Ashenafi Assefa [9,10], Awab Ghulam Rahim [11,12], Hoang Chau Nguyen[7], Tinh Hien Tran[7], Justin A. Green[13], Gavin C. K. W. Koh [14], Nicholas J. White[6,11], Nicholas Day[6,11], Dominic P. Kwiatkowski[1,16], Julian C. Rayner [15], Ric N. Price [2,6,11] & Sarah Auburn [2,6] ✉

[1]Wellcome Sanger Institute, Hinxton, Cambridge CB10 1SA, UK. [2]Menzies School of Health Research and Charles Darwin University, Darwin, Northern Territory 0811, Australia. [3]Eijkman Research Center for Molecular Biology, National Research and Innovation Agency, Jakarta 10430, Indonesia. [4]Institut Pasteur, University de Paris, Infectious Disease Epidemiology and Analytics Unit, Paris, France. [5]Exeins Health Initiative, Jakarta Selatan 12870, Indonesia. [6]Centre for Tropical Medicine and Global Health, Nuffield Department of Medicine, University of Oxford, Oxford OX3 7LJ, UK. [7]Oxford University Clinical Research Unit, Hospital for Tropical Diseases, 764 Vo Van Kiet, W.1, Dist.5, Ho Chi Minh City, Vietnam. [8]Department of Molecular Tropical Medicine and Genetics, Faculty of Tropical Medicine, Mahidol University, Bangkok, Thailand. [9]Ethiopian Public Health Institute, Addis Ababa, Ethiopia. [10]Institute for Global Health and Infectious Diseases, University of North Carolina at Chapel Hill, Chapel Hill, North Carolina, USA. [11]Mahidol-Oxford Tropical Medicine Research Unit, Faculty of Tropical Medicine, Mahidol University, Bangkok 10400, Thailand. [12]Afghan International Islamic University, Kabul, Afghanistan. [13]Formerly GlaxoSmithKline, Brentford, UK. [14]Department of Infectious Diseases, Northwick Park Hospital, Harrow, UK. [15]Cambridge Institute for Medical Research, University of Cambridge, Hills Road, Cambridge CB2 0XY, UK. [16]Deceased: Dominic P. Kwiatkowski. ✉e-mail: Sarah.Auburn@Menzies.edu.au

