## [Peer Review File · Nature Communications]

Lineage-informative microhaplotypes for recurrence classification and spatio-temporal surveillance of *Plasmodium vivax* malaria parasitesREVIEWER COMMENTS

Reviewer #1 (Remarks to the Author):

Authors here developed a panel of genome-wide 100 microhaplotypes, which exhibits high diversity among all the studied population across the globe. This developed panel showed equivalent results towards discrimination between relapse and reinfection *Plasmodium vivax* cases in comparison to genome-wide SNP barcodes studied earlier. The panel seems to be optimum for measurement of familial relatedness (Isolation by descent) and infer the lineages network among the isolates, which in turn can be explore for local infection outbreak and bottleneck events. In addition, the panel exhibited high accuracy in predicting the country of origin when compared to the genome-wide SNPs. The research article reported a new method of genome-wide genotyping which may require less genetic material and may also benefit in investigations of genetic differentiation between heterogenous parasites during management of effective usage of antimalarials and the spread of *P. vivax* in malaria endemic area. This article is highly recommended for publication with few clarifications on the below mentioned facts.

1. As spatio-temporal surveillance may face differential selection pressure. How local selection pressures can affect utility of these microhaplotypes for predicting the origin of the country/regional area can be a part of discussion.
2. Any observations of geographic region specific (country/cluster) microhaplotype diversity is not reported here. If such observation can also be discussed
3. Network among the isolates within/among country/s is not observed except in SAM panel 'a' as illustrated in Supplementary Figure 4. Regional microhaplotype-based infection networks. Does this mean, isolates from same country are not in infection network? For example, in case of 4- isolates from India in WAS are not in network in given panel 'c'. In addition, the panel 'b' for Afghanistan showed few isolates are in infection network but many not in a network while sampled from a single region. Does the distance between two isolates represented by two far apart circle means they are genetically differentiated. So, the non-cluster of the isolates within a country may be discussed. Can such network diagram for all the 615 isolates is possible to show a comprehensive global picture of infection network. A discussion on the observation gathered in this supplementary figure may be added.

Reviewer #2 (Remarks to the Author):

This manuscript from a group of researchers at several institutes including the Wellcome Trust Sanger Institute describes the development of 2 panels of genetic markers containing snps that can be used to determine relatedness between the malaria parasite *Plasmodium vivax* infections. This is important because vivax infections can be due to a relapse of the dormant liver stage, a recrudescence due to failed drug treatment, or a reinfection. Briefly, the team downloaded ~1800 publicly available Pv genomes from genome sequence databases, and identifying 1/3 of them (~615) as single genotypes, generated ~200 bp microhaplotype regions containing 3-10 snps that could be used to genotype Pv infections. The team generated 2 panels of 100 microhaplotypes (mhaps) called Random and High diversity, and compared against the Broad 42-snp panel. They first tested the 3 panels against simulated data and then in 14 pairs of infections whose relatedness was already determined through pre-existing data generated by others (whole genome sequencing). Not surprisingly, the panels worked better than the Broad panel since they contained almost 10-fold greater numbers of snps.

This is an interesting study but it falls short in my opinion of being worthy of a Nature Communications paper. A major concern is that the genomes that the authors mined are from only 17 countries of the 95 countries where Pv is endemic (Howe 2016). It's misleading to assign global regions when less than 20% of Pv endemic countries were sampled, especially since it's known that the genetic diversity of Pv makes country-specific genotypes highly likely. For example there are only 4 isolates included from India, and yet India is known to contribute enormously to the burden of Pv and to be highly genetically diverse. It's also unclear if the paper is introducing a new pipeline to be used by others as more Pv genome sequence data are generated; or a wet-lab method although few details are given about the amplicon-seq method to be able to use it in endemic countries.

Below are comments in no particular order:

1. Methods: "Illumina amplicon-based sequencing workflow as this methodology is widely used in the malaria community and has demonstrated utility and feasibility for malaria molecular surveillance in low- and middle-income countries¹⁷" the authors should cite the many other amplicon-seq studies that others in the malaria community pioneered instead of providing 1 recent reference that several of the authors on this paper have co-authored (Ref 17): for example the excellent studies by

- A. Lerch Sci Rep 2019
- K. Wamae JID 2022
- PN Rao JCM 2016
- E LaVerriere Mol Ecol 2022

2. The authors should add a sentence regarding the newly described endosplenic life cycle of Pv since infections are now being viewed as primarily infections of the spleen rather than exclusively as an infection of the blood. How will this influence panels of genetic markers and assays such as the one proposed that try to differentiate between relapse, reinfection and recrudescence?

3. Towards the end of the Results section, the authors introduce the recently published marker panels GEO33, GEO50 and GEO55 without having mentioned them previously in the paper almost as an afterthought. The GEO panels are not well described and the manuscript is unclear in its final conclusion regarding benchmarking against the GEO panels.

4. The authors need to clarify the statement in the Data availability section: Sequencing data from these samples has been made publicly available in the European Nucleotide Archive (ENA), with details provided in a data note describing the MalariaGEN Pv4.0 data set²⁰. Were any new data sets generated as part of this study? Or only mining of pre-existing datasets generated by others? The origin of the data should be explicitly given in the supplementary data Table 1 describing the samples.

6. Some of the statements concerning the prevalence of Pvivax need addressing:

- "Over the past decade, enhanced malaria control efforts in areas outside of sub-Saharan Africa have achieved a marked decline in *P. falciparum* infections, but a relative rise in the proportion of *P. vivax* cases¹." Is this correct? The citation is the WHO Malaria Report 2022, and this states "The proportion of cases due to *P. vivax* reduced from about 8% (20.5 million) in 2000 to 2% (4.9 million) in 2021." page xx which seems to contradict this statement.

- Is there "Increasing CQ drug resistance" in Pvivax? The reference provided is from 2014 and my understanding was that CQ resistance has not swept through Pv populations –providing a more recent reference(s) to back-up this statement would be useful

7. Table 1: please insert the names of the countries instead of the large global regions. For example, the authors use "Africa" to label the origin of the 47 isolates sequenced, but in fact they are all from Ethiopia which may not be representative of other countries in Africa (there are other Pv genomes sequenced from other countries in Africa available in public databases). In fact, the genomes that the authors mined are from only 17 countries of the 95 countries where Pv is endemic (Howe 2016). It's misleading to assign global regions when less than 20% of Pv endemic countries were sampled, especially since it's known that the genetic diversity of Pv makes country-specific genotypes highly likely.

Similarly the phrase "a global set of 615 *P. vivax* genomes" should be tempered to a set of 615 *P. vivax* genomes from 17 of the 95 endemic countries for Pv.

8. Fig 1. The authors describe the "core genome" being scanned, please provide a definition of the Pv core genome. Further in the same legend, the authors mention "microhaplotypes with the highest SNP densities tend to be located at the ends of the chromosomes" but how can this be if the core genome only was scanned? Still later is mentioned "Note, microhaplotypes were selected only from the accessible regions of the genome i.e., excluding highly diverse telomeric and sub-telomeric regions where sequence reads could not be mapped accurately." A definition of how the core genome was determined (see Otto et al 2018 for how the Pf core genome was delineated), and a better description of the methods regarding scanning and mapping would be useful. The term "accessible" is also misleading here, accessible to what? This is reminiscent of accessible chromatin used during studies of histone occupancy – which does not seem correct.

9. Fig 1d could be made more useful by including the centromeres and subtelomeric regions on the 14 chromosomes which might indicate what is present in the large gaps where no mhaps are mapped

Reviewer #3 (Remarks to the Author):

This is a review of Siegel et al's submission to Nature Communications, entitled "Lineage-informative microhaplotypes for spatio-temporal surveillance of Plasmodium vivax malaria parasites". Siegel and colleagues present a novel and promising genome-wide microhaplotype panel of 100 microhaplotypes capable of capturing *P. vivax* geographic diversity and predicting country of origin, but perhaps more importantly, capturing pairwise identity-by-descent (IBD) with comparable relatedness inferences to whole genome data and with higher accuracy than currently available bi-allelic SNP barcodes. This work addresses a persistent and critical issue in the vivax field – the ability to accurately genotype *P. vivax* infections to classify recurrent infections – with an approach that seems superior to current methods and that is amenable to large-scale, high-throughput amplicon sequencing platforms. Such an approach would undoubtedly pave the way for significant advances in our understanding of *P. vivax* epidemiology and biology, and especially to deepen our understanding of recurrences vs relapse in different endemic settings and in the context of therapeutic efficacy studies. The findings from the recurrent infection sample pairs are compelling and highlight the potential of this panel. Overall, this was a pleasure to read – the manuscript is well-written and will be of value to the community.

To further strengthen the results and overall impact, I believe the manuscript could be improved by attention to the following issues:

Major

Microhaplotype candidate selection and in silico validation: I found myself wondering in several instances of the manuscript how exactly the 100 microhaplotypes were chosen, given there were 5,460 candidates after the down-selection process. While I appreciate the authors describe the selection criteria (eg even spacing across the 14 chromosomes, high H_e , 3-10 SNPs), how many out of the 5,460 fit these criteria (surely this was not exactly 100)? Additionally, the authors describe the rationale for choosing 100 microhaps based on a modelling study (Taylor 2019) that showed this number of polyallelic markers was optimal for low RMSE considering the case of data-generating $r=0.5$ (it is worth noting that the Taylor study included only one *P. vivax* dataset and mostly *P. falciparum* data, although results were also based on simulated data). I don't find this rationale particularly compelling for the context of the present study.

Why weren't more than 100 markers considered to provide higher resolution? The microhap panel validation results highlight an important weakness, particularly that there is low accuracy in predicting sibling relationships ($r=0.5$) with very wide 95% CIs spanning 0.1-0.8, as well predicting country of origin for Cambodia and Vietnam (although perhaps this is unrelated to a marker panel resolution issue). Given the importance of achieving higher accuracy in predicting sibling relationships, and their implications for classifying recurrent infections in several scenarios, I would have thought a "sensitivity" analysis would have been independently carried out in the present study to determine an optimal marker number and whether including more than 100 microhaps may increase discriminatory resolution. Although the authors state, based on the Taylor study, that "Diminishing returns in RMSE reduction were observed above 100 polyallelic markers, highlighting this target number as a pragmatic balance of IBD accuracy against the economic cost of primers", I still think there is a need for validation (or at a minimum confirmation) of whether this number is indeed optimal for the *P. vivax* microhap panel.

Data simulations: I appreciate the authors acknowledgment of the limitations of the simulated data in the discussion. However, given the importance of the simulated data for validation of pairwise IBD estimation in this study, I find the arguments a bit weak and some of these limitations may be quite important. Was there a reason why they were not included (eg miss-specifications, genotyping errors, missing data etc)? What was the rationale for simulating 100 haploid genotype pairs? This information should be provided. I think the elephant in the room is also how well the panel would perform in the case where infections are $COI > 1$. It could be possible to explore this with simulation (at least up to $COI = 2$) to enable an assessment of the (imperfect) detection of minority clones and limitations to IBD estimation. At a minimum this should be

expanded upon in the discussion as this will surely continue to be a key issue going forward for any microhap or barcoding panel. Related to my comment above, it seems that an iterative process could have been employed to evaluate a range of microhap numbers (eg 100, 125, 150) in the panel and their performance assessed quite easily with paneljudge.

Reproducibility/code availability: Given this in silico study is intended to provide a flexible bioinformatics pipeline for the vivax community, the pipeline itself and description of the Github repository can still be clearer and more user-friendly (eg README, etc). Indeed, the authors state in the discussion "However, the informatics pipeline that was established for the microhaplotype selection can be applied readily to update the panel as needed once additional whole genome data become available from new geographical regions. The pipeline can also be used to select country- or region-specific panels where needed." I agree that the authors have developed a robust framework for microhaplotype selection that will be useful for the described scenarios, however, additional improvements are required to make this more accessible to users. There are also no scripts provided for the analyses presented in this manuscript, although they are described in Methods. This is certainly the expectation for in silico studies and computational analyses and these scripts should also be made available for reproducibility.

Minor comments

- Title/abstract/overall: while the title/abstract focus more on the spatial features (geographic diversity, prediction of country of origin) captured by the microhap panel, the rest of the manuscript focuses (particularly the introduction and discussion somewhat) on the need for a tool/approach that captures pairwise IBD inferences and enables recurrence classification. Seems a bit unbalanced
- Intro: given the panel was really derived from the 615 genomes (not 1816) it seems slightly confusing to include this number in the introductory text
- Figure 2: the legend could be more informative to aid the reader (avoid needing to refer to text)
- Table 4 ref: I believe the authors should reference Table 3, not Table 4 in the following, "The highest effective cardinality observed in the data set was a microhaplotype with a score of 36.9 (roughly 36 different alleles at a single microhaplotype marker) in West Southeast Asia (Table 4)."
- Spatial differentiation patterns (Fig 5): I agree with the authors that "the microhaplotype-panel based PCoA trends were consistent with spatial trends observed with genome-wide datasets", however, it should be acknowledged that differentiation between Oceania/MSEA and ESEA/WSEA was less marked. Again I wonder if inclusion of additional key microhaps with high H_e in these regions may increase discriminatory resolution? There would perhaps be a trade-off, but did the authors consider this?
- Pairwise IBD (Supp Fig 3): Although Supp Fig 3 shows significant positive correlations between microhap-based IBD and genome-wide IBD in all geographic regions, there does appear to be a trend for 'misclassification' of IBD when genome-wide IBD is close to 0, with a range of microhap IBD from 0~0.3. Do the authors have any speculation what might be driving this? It seems this could lead to an important misclassification of "strangers" to "distant relatives" and perhaps "close relatives". Apologies if this is just my misinterpretation of the plot (admittedly it is a bit hard to see – maybe adding an alpha transparency to the points may help)
- Supp Fig 4: I think there is an error in interpretation or an oversight with respect to the cross-border transmission networks. They occur between Cambodia and Vietnam, not Thailand. Also, there is repeated text in the legend

Response to reviewers

Reviewer #1 (Remarks to the Author):

Authors here developed a panel of genome-wide 100 microhaplotypes, which exhibits high diversity among all the studied population across the globe. This developed panel showed equivalent results towards discrimination between relapse and reinfection *Plasmodium vivax* cases in comparison to genome-wide SNP barcodes studied earlier. The panel seems to be optimum for measurement of familial relatedness (Isolation by descent) and infer the lineages network among the isolates, which in turn can be explore for local infection outbreak and bottleneck events. In addition, the panel exhibited high accuracy in predicting the country of origin when compared to the genome-wide SNPs. The research article reported a new method of genome-wide genotyping which may require less genetic material and may also benefit in investigations of genetic differentiation between heterogenous parasites during management of effective usage of antimalarials and the spread of *P. vivax* in malaria endemic area. This article is highly recommended for publication with few clarifications on the below mentioned facts.

Response: We thank the reviewer for their positive feedback.

1. As spatio-temporal surveillance may face differential selection pressure. How local selection pressures can affect utility of these microhaplotypes for predicting the origin of the country/regional area can be a part of discussion.

Response: We have revised the discussion to include this point as follows:

Lines 451-453: *"It is possible that some microhaplotypes are subject to site-specific selective pressure, which could affect their spatio-temporal surveillance utility; this impact could be mitigated by implementing additional microhaplotypes."*

2. Any observations of geographic region specific (country/cluster) microhaplotype diversity is not reported here. If such observation can also be discussed

Response: As detailed in the Methods, the high diversity microhaplotypes were intentionally selected to exhibit high diversity in each of the 7 geographic regions assessed. Please find plots illustrating the marker diversity and effective cardinality in each geographic region presented in **Figures 4 a and b**. We also note that the high diversity 100 microhaplotype panel is only an example, and one can select alternative markers to fit specific use cases.

3. Network among the isolates within/among country/s is not observed except in SAM panel 'a' as illustrated in Supplementary Figure 4. Regional microhaplotype-based infection networks. Does this mean, isolates from same country are not in infection network? For example, in case of 4-isolates from

India in WAS are not in network in given panel 'c'. In addition, the panel 'b' for Afghanistan showed few isolates are in infection network but many not in a network while sampled from a single region. Does the distance between two isolates represented by two far apart circle means they are genetically differentiated. So, the non-cluster of the isolates within a country may be discussed.

Response: We apologize for the confusion. To clarify this, we now provide additional information in the legend of **Supplementary Figure 4 (lines 880-883)** which states *"Where two circles are not connected by a line, the estimated IBD between those infections was below the given threshold of 0.25 (i.e. they were not inferred to be highly related). Alternative IBD thresholds can be applied to detect more distant connectivity between infections. The distance between infections (circles) that are not connected by lines does not reflect the relatedness between those infections"*.

Can such network diagram for all the 615 isolates is possible to show a comprehensive global picture of infection network. A discussion on the observation gathered in this supplementary figure may be added.

Response: The accuracy of the IBD estimations (that are used to construct the networks) depends in part on the population allele frequency information provided. For a global network analysis across all 615 samples, a globally averaged allele frequency input would be required, which might compromise the accuracy of many IBD estimations; we therefore did not attempt this analysis. **Lines 280-281** describe this rationale as follows *"To retain accuracy in MAF estimates, IBD-based analyses were restricted to within each of the 7 regional groupings."*

Reviewer #2 (Remarks to the Author):

This manuscript from a group of researchers at several institutes including the Wellcome Trust Sanger Institute describes the development of 2 panels of genetic markers containing snps that can be used to determine relatedness between the malaria parasite *Plasmodium vivax* infections. This is important because *vivax* infections can be due to a relapse of the dormant liver stage, a recrudescence due to failed drug treatment, or a reinfection. Briefly, the team downloaded ~1800 publicly available Pv genomes from genome sequence databases, and identifying 1/3 of them (~615) as single genotypes, generated ~200 bp microhaplotype regions containing 3-10 snps that could be used to genotype Pv infections. The team generated 2 panels of 100 microhaplotypes (mhaps) called Random and High diversity, and compared against the Broad 42-snp panel. They first tested the 3 panels against simulated data and then in 14 pairs of infections whose relatedness was already determined through pre-existing data generated by others (whole genome sequencing). Not surprisingly, the panels worked better than the Broad panel since they contained almost 10-fold greater numbers of snps.

This is an interesting study but it falls short in my opinion of being worthy of a Nature Communications paper. A major concern is that the genomes that the authors mined are from only 17 countries of the 95 countries where Pv is endemic (Howe 2016). It's misleading to assign global regions when less than 20% of Pv endemic countries were sampled, especially since it's known that the genetic diversity of Pv makes

country-specific genotypes highly likely. For example there are only 4 isolates included from India, and yet India is known to contribute enormously to the burden of Pv and to be highly genetically diverse. It's also unclear if the paper is introducing a new pipeline to be used by others as more Pv genome sequence data are generated; or a wet-lab method although few details are given about the amplicon-seq method to be able to use it in endemic countries.

Response: We thank the reviewer for their feedback, including their acknowledgment of the importance of recurrence classification for *P. vivax*. A brief reply addressing each of their concerns is provided below.

Firstly, we add clarity to the manuscript in response to the confusion about whether our study aimed to present *in silico* as well as laboratory-based outputs. We confirm that our study is purely *in silico*. The primary aims were to establish and to evaluate the accuracy of *P. vivax* microhaplotype panels in capturing relatedness (as measured by identity by descent) between infection pairs. To avoid any confusion to other readers, we have clarified several sections of the text, including the following lines in the abstract:

- **Lines 46-49**, “We have developed a *P. vivax* marker discovery framework to identify and select panels of microhaplotypes (multi-allelic markers within small, amplifiable segments of the genome) that can accurately capture IBD. Using a global set of 615 *P. vivax* genomes, we discovered and evaluated panels of 50-250 microhaplotypes.”
- **Lines 55-56**, “Our framework is available open-source for users to discover and select microhaplotypes customised to their needs, with potential for porting to other species or data resources.”

The reviewer questions why only 17 countries were used for the study and whether this is representative of the global *P. vivax* burden. There are two important considerations here; 1) understanding of the global epidemiology of *P. vivax*, and 2) understanding of the constraints in the available genomic data on *P. vivax*.

Regarding the first consideration, the reviewer suggests that there are 95 vivax-endemic countries, referencing a 2016 study (using 2010 data) by the Malaria Atlas Project (Howes et al., Am J Trop Med Hyg 2016). This is not an accurate description of the current global distribution of *P. vivax* endemicity. More recent estimates indicate that there are 49 countries with stable and quantifiable levels of *P. vivax* transmission (Price et al., Trends Parasitol 2020; World Health Organisation 2022 World Malaria Report). Furthermore, *P. vivax* incidence data generated by the Malaria Atlas Project in 2020 reports incidence >0 in 42 countries (<https://malariaatlas.org/>). The 17 countries included in our original (Pv4.0) analysis therefore represent >40% of the global distribution of *P. vivax*. If we consider the newly investigated independent (non-Pv4.0) dataset we have now included for validation purposes, our assessments cover 26 countries, representing >50% of the global distribution.

Regarding the second consideration, the reviewer may not be aware of the immense challenges in generating genomic data on *P. vivax*. The authors of our study are very aware of these challenges as many of us have worked within the vivax Genomic Epidemiology Network (vivaxGEN), which generated >50% of the *P. vivax* genomes in the MalariaGEN Pv4 dataset that was utilized for this study; it took a

decade of work with vivax-endemic partners from 16 countries to generate this genomic data and represents the vast majority of publicly available data to date. Genomic data from clinical *P. vivax* isolates are substantially harder to generate than that for *P. falciparum* due to significantly lower parasite densities in the circulating blood of infected individuals and the inability to maintain *P. vivax* in continuous *ex vivo* culture. These challenges underlie our rationale for developing a framework to select markers for targeted genotyping (please see lines 116-123). We originally used the MalariaGEN Pv4 dataset as this represents the largest collated collection of high-quality genomes for *P. vivax*. In response to the reviewer request, we have since retrieved all the published, openly accessible *P. vivax* genomes that we could find from other sources and used this as an independent external validation dataset. We have added a new section to the manuscript describing the independent validation data and evaluation (detailed methods in Supplementary Document 1, and results in lines 305-322). From an initial set of 836 non-Pv4 genomes, after quality filtering and removal of polyclonal infections, we retrieved an additional 288 new *P. vivax* genomes for evaluation of an exemplar 100-microhaplotype panel selected from the Pv4 dataset. In brief, significant correlation was observed between the 100-microhaplotype panel and genome-wide data in the non-Pv4 data, confirming accuracy in IBD estimation in this independent external validation set (see Supplementary Figure 6). As such we believe that our dataset provides a solid basis for exploring globally applicable microhaplotypes. We thank the reviewers for this suggestion, as we believe the newly added analysis considerably strengthens the value proposition of this work by demonstrating that this framework and findings are extensible beyond the initial Pv4 data, and that the candidate microhaplotype panels we evaluated are robust against the addition of data from poorly characterized geographic regions of the world.

Below are comments in no particular order:

1. Methods: “Illumina amplicon-based sequencing workflow as this methodology is widely used in the malaria community and has demonstrated utility and feasibility for malaria molecular surveillance in low- and middle-income countries¹⁷” the authors should cite the many other amplicon-seq studies that others in the malaria community pioneered instead of providing 1 recent reference that several of the authors on this paper have co-authored (Ref 17): for example the excellent studies by

- A. Lerch Sci Rep 2019
- K. Wamae JID 2022
- PN Rao JCM 2016
- E LaVerriere Mol Ecol 2022

Response: This was an unintentional omission and agree that these other studies are important to include here. The Jacobs et al eLife 2021 (ref 17) is one of the few studies that has generated amplicon sequencing data on malaria (in this case *P. falciparum*) that is being used by National Malaria Control Programs to inform policy and practice (see the article’s description of treatment policy changes in Vietnam). We have added the new references (refs 18-20).

2. The authors should add a sentence regarding the newly described endosplenic life cycle of Pv since infections are now being viewed as primarily infections of the spleen rather than exclusively as an

infection of the blood. How will this influence panels of genetic markers and assays such as the one proposed that try to differentiate between relapse, reinfection and recrudescence?

Response: We agree that the endosplenic life cycle of *P. vivax* is important and warrants further investigation. We are working actively with colleagues (at Menzies) leading these splenic studies to explore this area. In view of the challenge of obtaining high-quality *P. vivax* genomes from splenic tissue, we plan to apply microhaplotype genotyping for this purpose (NHMRC grant 2019153). To answer the specific question raised by the reviewer (and any other interested readers), we have added some comments to the discussion on **lines 425-431**:

*“A recent study highlights the human spleen as an important reservoir of *P. vivax* parasites. However, there is currently no evidence of altered metabolism of endosplenic stages in human *P. vivax* infections, in marked contrast to the hypnozoite reservoir. We therefore anticipate that the endosplenic stages are treatable by blood-stage antimalarials such as chloroquine, although this needs to be confirmed. In this context, information on IBD between paired clinical isolates would not distinguish between endosplenic and circulating infections but should still help to distinguish reactivated liver stages and reinfections.”*

3. Towards the end of the Results section, the authors introduce the recently published marker panels GEO33, GEO50 and GEO55 without having mentioned them previously in the paper almost as an afterthought. The GEO panels are not well described and the manuscript is unclear in its final conclusion regarding benchmarking against the GEO panels.

Response: We have also revised the text in the Discussion (**lines 440-443**) to give the reader more context on the microhaplotype results relative to the GEO panels as follows:

“The equivalently high prediction accuracy of the microhaplotypes relative to the GEO panels means that users who want to capture information on both IBD and on country of origin can derive all information with the microhaplotypes alone without impeding on accuracy”.

4. The authors need to clarify the statement in the Data availability section:

Sequencing data from these samples has been made publicly available in the European Nucleotide Archive (ENA), with details provided in a data note describing the MalariaGEN Pv4.0 data set²⁰. Were any new data sets generated as part of this study? Or only mining of pre-existing datasets generated by others? The origin of the data should be explicitly given in the supplementary data Table 1 describing the samples.

Response: We have added further details to the data availability section, ensuring everything needed to replicate the study results are provided. See lines **677-683**:

*“Sequencing data from the MalariaGEN Pv4 samples has been made publicly available in the European Nucleotide Archive (ENA), with details provided in a data resource describing the MalariaGEN Pv4 data²⁵. Accession codes are also available in **Supplementary Table 1** and the VCF can be downloaded here: <https://www.malariagen.net/resource/30>. The sequencing data from the independent validation samples has also been made publicly available in the ENA or National Institutes of Health Sequence*

Repository Archive in the respective contributing studies^{32–35}. Accession codes are also available in **Supplementary Table 1.**”

We also emphasize that, although the work was based on previously sequenced samples published in Pv4, ~50% of the data in Pv4 were generated by the vivax Genomic Epidemiology Network (vivaxGEN), which was established by authors Auburn and Price (see sample and partner attributions in Pv4 publication). In addition, many of the co-authors of this study were directly involved in clinical studies that established patient sampling (Assefa, Rahim, Chau, Hien, Green, Koh, White, Day and Price). Furthermore, Kwiatkowski (founder of MalariaGEN) and Rayner teams at Wellcome Sanger Institute (including Siegel, Amato, Murie and Whitton) conducted the whole genome sequencing that generated and analyzed ~86% of the data in Pv4 and conducted the mapping and variant calling to produce the genotyping data for all samples in the Pv4 data resource. It is therefore inaccurate to describe our work as “mining of pre-existing datasets generated by others”. Rather, many of the co-authors have actively generated the majority of the data used in this study. Additionally, Pv4 was the first study of its kind to create a unified data resource both from in-house generated data, as well as external data that was available in the wider community (at that point in time) to make data easier to analyze for the entire malaria community.

5. Some of the statements concerning the prevalence of Pvivax need addressing:

5a. “Over the past decade, enhanced malaria control efforts in areas **outside of sub-Saharan Africa** have achieved a marked decline in *P. falciparum* infections, but a relative rise in the proportion of *P. vivax* cases1.” Is this correct? The citation is the WHO Malaria Report 2022, and this states “The proportion of cases due to *P. vivax* reduced from about 8% (20.5 million) in 2000 to 2% (4.9 million) in 2021.” page xx which seems to contradict this statement.

Response: The reviewer is referring to the global burden of malaria which has been dominated by an upsurge in the prevalence of *P. falciparum*. The statement in the text quoted explicitly refers to “areas outside of Africa”, where *P. falciparum* and *P. vivax* are co-endemic and the relative *P. vivax* burden has risen (Price Trends in Parasitology 2020). This important observation highlights the relatively greater challenges of eliminating a parasite capable of relapsing weeks to months after the initial infection.

5b. Is there “Increasing CQ drug resistance” in Pvivax? The reference provided is from 2014 and my understanding was that CQ resistance has not swept through Pv populations –providing a more recent reference(s) to back-up this statement would be useful

Response: This is an important question and one that has been hard to answer. Our 2014 review (Price et al Lancet 2014) was undertaken more than 9 years ago. Whilst there is evidence of low grade CQR in many locations, the main areas of high-grade resistance are in Sabah (Malaysia), Papua (Indonesia) and Papua New Guinea. Some areas have already changed to Artemisinin Combination Therapy (ACT). Ex vivo drug susceptibility testing is confounded by an inability to maintain *P. vivax* in continuous ex vivo

culture and, in areas where CQ is still used, recrudescence is confounded by reinfection and relapse, confounding clinical definitions of CQR. These challenges are a key reason for developing the microhaplotype methodology to discriminate between relapses, reinfections and recrudescence.

6. Table 1: please insert the names of the countries instead of the large global regions. For example, the authors use “Africa” to label the origin of the 47 isolates sequenced, but in fact they are all from Ethiopia which may not be representative of other countries in Africa (there are other Pv genomes sequenced from other countries in Africa available in public databases). In fact, the genomes that the authors mined are from only 17 countries of the 95 countries where Pv is endemic (Howe 2016). It’s misleading to assign global regions when less than 20% of Pv endemic countries were sampled, especially since it’s known that the genetic diversity of Pv makes country-specific genotypes highly likely.

Response: We have added country names to **Supplementary Table 1** to provide information on the countries included in the study. However, we have refrained from using country names to classify the groupings of clinical isolates for several reasons that we have clarified in the Methods (lines 520-527) as follows: *“Rather than using country-based classifications to define infection groups, we employed the regional classifications defined in the MalariaGEN Pv4 data resource, which are based on both geographic location and genomic clustering patterns. This approach is more accurate than purely country-based classifications since malaria transmission networks do not always conform to national spatial boundaries. For example, there is a malaria-free ‘corridor’ that runs through the middle of Thailand, leading to distinct separation of eastern and western infections⁴⁴. In contrast, some border regions, such as parts of Vietnam and Cambodia, are very ‘porous’ with high levels of homology seen between infections across national borders.”*

7. Similarly the phrase “a global set of 615 P. vivax genomes” should be tempered to a set of 615 P. vivax genomes from 17 of the 95 endemic countries for Pv.

Response: We would prefer to retain our original wording as all 7 of the broadly defined vivax endemic geographical sub-regions (i.e. global) are represented (for details please see Pv4.0 data resource for how these were classified, and how many samples and countries were represented in each geographical subpopulation). As the geographic classification/assignment for each sample is based on analysis of the wider Pv4 dataset, which comprised 1,895 samples from 27 countries, and not 615 samples from 17 countries, one can consider the 615 highest quality monoclonal samples used as a subset of those found in Pv4.0. The reference to 95 vivax-endemic countries is inaccurate (please see our earlier response).

8. Fig 1. The authors describe the “core genome” being scanned, please provide a definition of the Pv core genome. Further in the same legend, the authors mention “microhaplotypes with the highest SNP densities tend to be located at the ends of the chromosomes” but how can this be if the core genome only was scanned? Still later is mentioned “Note, microhaplotypes were selected only from the accessible regions of the genome i.e., excluding highly diverse telomeric and sub-telomeric regions where sequence reads could not be mapped accurately.” A definition of how the core genome was

determined (see Otto et al 2018 for how the Pf core genome was delineated), and a better description of the methods regarding scanning and mapping would be useful. The term “accessible” is also misleading here, accessible to what? This is reminiscent of accessible chromatin used during studies of histone occupancy – which does not seem correct.

9. Fig 1d could be made more useful by including the centromeres and subtelomeric regions on the 14 chromosomes which might indicate what is present in the large gaps where no mhaps are mapped

Response (to 8 and 9): We acknowledge that the nomenclature could be misleading as these terms have different connotations in other fields, and is not aided by the fact that “core” and “accessible” genome are commonly used interchangeably. In this context, accessibility refers to the ability to confidently discover and genotype variants in a region given sequence data and methods comparable to those used in the original studies (short read sequencing, in particular). This definition of accessibility is commonly used also in other species (see DOI 10.1038/nature24995 for an example in mosquito genomics). Here we have used the definition of core genome provided as part of the MalariaGEN Pv4 data package, which in turn extends the one in Pearson et al. 2016 and is based on systematically low coverage and mapping quality across all samples in the set (hence more likely due to the intrinsic “accessibility” of the genome with short reads). While we understand the potential for confusion, on balance we believe that the most transparent approach is to retain the nomenclature used in the Pv4 data release and in the manuscript. Changes to the manuscript for clarity include:

- 1) We have removed the word ‘accessible’ throughout the text and simply used the phrase ‘core genome regions’. See further detail here as well in lines 772-775: *“The microhaplotypes with the highest SNP densities tend to be located at the ends of the chromosomes. Note, microhaplotypes were selected only from the core regions of the genome i.e., excluding highly diverse telomeric, and sub-telomeric and centromere regions where sequence reads could not be mapped accurately (see Pv4 data resource for further details on core regions)”*.
- 2) Additionally, we have added a visual of the proximity of markers along the genome along with identifying the non-core genome boundaries in the marker panel discovery framework (jupyter notebook 2 and Figure 2a) so that you can see where these regions lie and where your selected marker panels are in relation to them. From this analysis in notebook 2, you can tell that microhaplotypes are indeed more common along the ends of chromosomes (i.e. near the core and non-core boundaries), which are an indication that the boundaries of centromeres and telomeres as currently defined by Pv4 are likely imperfect based on this pronounced effect. We expect the high density of markers at the ends of chromosomes is an artefact of imperfect core region boundaries as previously defined in Pv4/Pv1 from the original reference genome characterisation, but we feel that stating this in the manuscript doesn’t contribute to value add in this case and also would be difficult to conclusively demonstrate.

Reviewer #3 (Remarks to the Author):

This is a review of Siegel et al's submission to Nature Communications, entitled "Lineage-informative microhaplotypes for spatio-temporal surveillance of Plasmodium vivax malaria parasites". Siegel and colleagues present a novel and promising genome-wide microhaplotype panel of 100 microhaplotypes capable of capturing *P. vivax* geographic diversity and predicting country of origin, but perhaps more importantly, capturing pairwise identity-by-descent (IBD) with comparable relatedness inferences to whole genome data and with higher accuracy than currently available bi-allelic SNP barcodes. This work addresses a persistent and critical issue in the vivax field – the ability to accurately genotype *P. vivax* infections to classify recurrent infections – with an approach that seems superior to current methods and that is amenable to large-scale, high-throughput amplicon sequencing platforms. Such an approach would undoubtedly pave the way for significant advances in our understanding of *P. vivax* epidemiology and biology, and especially to deepen our understanding of recurrences vs relapse in different endemic settings and in the context of therapeutic efficacy studies. The findings from the recurrent infection sample pairs are compelling and highlight the potential of this panel. Overall, this was a pleasure to read – the manuscript is well-written and will be of value to the community.

Response: We thank the reviewer for acknowledging the importance that our study brings to the *P. vivax* field relative to the current literature.

To further strengthen the results and overall impact, I believe the manuscript could be improved by attention to the following issues:

Major

1. Microhaplotype candidate selection and in silico validation: I found myself wondering in several instances of the manuscript how exactly the 100 microhaplotypes were chosen, given there were 5,460 candidates after the down-selection process. While I appreciate the authors describe the selection criteria (eg even spacing across the 14 chromosomes, high H_e , 3-10 SNPs), how many out of the 5,460 fit these criteria (surely this was not exactly 100)?

Response: We fully agree that this is not as clear as it could have been. Therefore we have now added more details and commentary to the microhaplotype marker discovery framework as part of this manuscript (see mentions to github repo from vsiegel throughout the manuscript) that walks users through options to conduct their own marker and panel discovery in a fully customisable set of jupyter notebooks, which we also used to help select the "high-diversity" and "random" marker sets that are included in the paper. These notebooks are now directly linked to the Pv4 cloud-based data resource (no need to download anything - and just a few clicks to analyze it yourself). To add clarity to the methods, we have broadened the framework and its accompanying notebooks to include less hard filtering (therefore it is more customisable) and included the following new text: "We broadly defined the Pv heterozygote as all microhaplotype window selections with at least 2 SNPs and heterozygosity ≥ 0.5 , but these thresholds are arbitrary.

Panel selection. We began panel selection by dividing the total length of the 14 chromosomes by 100 (parameter that is customisable), to assign the proportional number of markers needed per chromosome. The main question remaining on optimisation at this stage was to evaluate how important marker spacing was compared to heterozygosity, when analysed in reference to a traditional SNP-based panel. Therefore, we wrote two algorithms to generate panels that either prioritised spacing (see notebook 2 algorithm called “evenly-spaced”), or heterozygosity (see notebook 2 algorithm called “greedy”) as a first attempt to generate the high-diversity and random panels. However, as both optimisation algorithms are limited in their ability to simultaneously optimise on both heterozygosity and marker spacing, the final panel selection stage for the high-diversity and random panels was done manually by visual inspection (from outputs of the “greedy” and “evenly-spaced” algorithms, respectively). They were then hard filtered with the added stringency of needing at least 3 SNPs and heterozygosity ≥ 0.6 (total windows across the whole genome = 1,627).

Discussion. The framework we include allows the user to choose what is more important to prioritize - the spacing of the individual markers, or the marker’s diversity, providing two versions of this kind of optimisation and selection. There is no right answer to this question, it is ultimately up to the user to decide which matters more to them. The selection of the two candidate panels were designed to explore optimisation of heterozygosity and spacing, which was performed using the discovery framework and analysis notebooks, including dividing the chromosomes into equal sized sections, and then choosing each marker set manually based on outputs of the two algorithms (greedy or evenly spaced), and then a final manual selection trying to optimize both criteria of spacing and diversity (high-diversity panel), or randomly selected on spacing criteria alone without explicit optimisation of diversity (random panel, although minimum heterozygosity > 0.6). While the two algorithms detailed are semi-automated, users may want to do the final selection themselves, as we have done, or simply generate a candidate panel automatically from one of the two algorithms. We should emphasize that the two candidate panels we described initially are good representative examples of panels that could be used and would perform in a superior way to a SNP panels such as BR38. The two panels would not be considered the “ideal” panel, as there really is no such thing as ideal in absolute terms, simply different choices of prioritization.

2. Additionally, the authors describe the rationale for choosing 100 microhaps based on a modelling study (Taylor 2019) that showed this number of polyallelic markers was optimal for low RMSE considering the case of data-generating $r=0.5$ (it is worth noting that the Taylor study included only one *P. vivax* dataset and mostly *P. falciparum* data, although results were also based on simulated data). I don’t find this rationale particularly compelling for the context of the present study.

3. Why weren’t more than 100 markers considered to provide higher resolution? The microhap panel validation results highlight an important weakness, particularly that there is low accuracy in predicting sibling relationships ($r=0.5$) with very wide 95% CIs spanning 0.1-0.8, as well predicting country of origin for Cambodia and Vietnam (although perhaps this is unrelated to a marker panel resolution issue). Given the importance of achieving higher accuracy in predicting sibling relationships, and their implications for classifying recurrent infections in several scenarios, I would have thought a “sensitivity” analysis would

have been independently carried out in the present study to determine an optimal marker number and whether including more than 100 microhaps may increase discriminatory resolution. Although the authors state, based on the Taylor study, that “Diminishing returns in RMSE reduction were observed above 100 polyallelic markers, highlighting this target number as a pragmatic balance of IBD accuracy against the economic cost of primers”, I still think there is a need for validation (or at a minimum confirmation) of whether this number is indeed optimal for the *P. vivax* microhap panel.

Response (to 2 and 3): We note that the microhaplotype marker discovery framework allows the user to define as many markers as they choose for their panels. We have now extended our analyses to include panels of 50-250 microhaplotypes. The results of these analyses are described in lines 325-344 and illustrated in Supplementary Figure 7. In brief, as previously demonstrated by Taylor and colleagues (Taylor *et al.*, *Genetics*, 2019), we find that “increasing panel size improves the accuracy of IBD estimation, with a large gain in panel informativeness between 50 and 100 microhaplotypes but diminishing returns above 100 microhaplotypes”. As we describe in the Discussion (lines 358-361) “The marginal gains to informativeness at higher panel sizes (150, 200, 250) are consistent with modelling studies by Taylor and colleagues and come at a trade-off in cost and laboratory practicalities. Panels of 100 microhaplotypes provide a reasonable compromise.”

4. Data simulations: I appreciate the authors acknowledgment of the limitations of the simulated data in the discussion. However, given the importance of the simulated data for validation of pairwise IBD estimation in this study, I find the arguments a bit weak and some of these limitations may be quite important. Was there a reason why they were not included (eg miss-specifications, genotyping errors, missing data etc)? What was the rationale for simulating 100 haploid genotype pairs? This information should be provided.

Response: We were unfortunately not able to add simulations with missing data as this would require substantial reprogramming of the *paneljudge* software, which is beyond the remit of this study. We have clarified this challenge in the Discussion (lines 383-386):
“... it should be acknowledged that genotyping errors and failures (missing data) were not included in our simulations as this would require reprogramming of the *paneljudge* package, which is beyond the remit of the current study.”

We should also note that we obtained some (albeit non-systematic) insight into the impact of missing data within our evaluation of IBD estimation at a 100 microhaplotype panel using ‘real’ data within the independent (non-Pv4) validation dataset. The samples within the independent dataset exhibited up to 50% missing data but we did not find any evidence that the samples with greatest proportion of missing data exhibited greatest deviation in IBD estimation relative to the genome-wide ‘gold standard’.

5. I think the elephant in the room is also how well the panel would perform in the case where infections are $COI > 1$. It could be possible to explore this with simulation (at least up to $COI = 2$) to enable an assessment of the (imperfect) detection of minority clones and limitations to IBD estimation. At a minimum this should be expanded upon in the discussion as this will surely continue to be a key issue going forward for any microhap or barcoding panel. Related to my comment above, it seems that an

iterative process could have been employed to evaluate a range of microhap numbers (eg 100, 125, 150) in the panel and their performance assessed quite easily with paneljudge.

Response: The COI query is certainly an area of interest but would require substantial reprogramming of the *paneljudge* software, which is not in the remit of this study. We have however integrated new data on a larger range of microhaplotype panels as described above.

6. Reproducibility/code availability: Given this in silico study is intended to provide a flexible bioinformatics pipeline for the vivax community, the pipeline itself and description of the Github repository can still be clearer and more user-friendly (eg README, etc). Indeed, the authors state in the discussion “However, the informatics pipeline that was established for the microhaplotype selection can be applied readily to update the panel as needed once additional whole genome data become available from new geographical regions. The pipeline can also be used to select country- or region-specific panels where needed.” I agree that the authors have developed a robust framework for microhaplotype selection that will be useful for the described scenarios, however, additional improvements are required to make this more accessible to users. There are also no scripts provided for the analyses presented in this manuscript, although they are described in Methods. This is certainly the expectation for in silico studies and computational analyses and these scripts should also be made available for reproducibility.

Response: We agree that additional details were needed to make the repo more intuitive and user friendly, and now have added substantial detail to the microhaplotype marker discovery framework, including the annotation of the full end-to-end discovery pipeline used in our analysis, as well as additional code for exploring and generating candidate panels that could be potentially useful to users (which are fully customisable and can be optimized based on the user’s desires and prioritization needs). We hope that the significant rework to the github repo has addressed these concerns. Please see <https://github.com/svsiegel/vivax-mhaps> for the new version.

Minor comments

- Title/abstract/overall: while the title/abstract focus more on the spatial features (geographic diversity, prediction of country of origin) captured by the microhap panel, the rest of the manuscript focuses (particularly the introduction and discussion somewhat) on the need for a tool/approach that captures pairwise IBD inferences and enables recurrence classification. Seems a bit unbalanced

Response: We have revised the title to “**Lineage-informative microhaplotypes for recurrence classification and spatio-temporal surveillance of *Plasmodium vivax* malaria parasites**”

- Intro: given the panel was really derived from the 615 genomes (not 1816) it seems slightly confusing to include this number in the introductory text

Response: We have revised this to 615 in the abstract (line 49) and introduction (line 134).

- Figure 2: the legend could be more informative to aid the reader (avoid needing to refer to text)

Response: We have updated this (lines 788-793): *“Confidence intervals (CI, shaded regions) are based on data simulated using 5 data-generating relatedness parameters, r . Data are presented on 3 marker panels: High-diversity microhaplotype panel (purple), Random-SNP microhaplotype panel (orange), and 38 Broad barcode biallelic SNPs (green). Separate plots are provided for each r (from 0.01-0.99) and geographic region; AF (Africa), ESEA (East Southeast Asia), MSEA (Maritime Southeast Asia), OCE (Oceania), SAM (South America), WAS (West Asia) and WSEA (West Southeast Asia).”*

- Table 4 ref: I believe the authors should reference Table 3, not Table 4 in the following, “The highest effective cardinality observed in the data set was a microhaplotype with a score of 36.9 (roughly 36 different alleles at a single microhaplotype marker) in West Southeast Asia (Table 4).”

Response: Thank you for noting this. We have removed this table to reduce emphasis on specific panels.

- Spatial differentiation patterns (Fig 5): I agree with the authors that “the microhaplotype-panel based PCoA trends were consistent with spatial trends observed with genome-wide datasets”, however, it should be acknowledged that differentiation between Oceania/MSEA and ESEA/WSEA was less marked. Again I wonder if inclusion of additional key microhaps with high H_e in these regions may increase discriminatory resolution? There would perhaps be a trade-off, but did the authors consider this?

Response: We agree that users could and certainly should append their microhaplotype panels as needed for specific use cases. We have clarified this message in a prominent section of the manuscript (last sentence of the abstract, lines 55-57) as follows: *“Our framework is available open-source for users to discover and select microhaplotypes customised to their needs, with potential for porting to other species or data resources.”*

- Pairwise IBD (Supp Fig 3): Although Supp Fig 3 shows significant positive correlations between microhap-based IBD and genome-wide IBD in all geographic regions, there does appear to be a trend for ‘misclassification’ of IBD when genome-wide IBD is close to 0, with a range of microhap IBD from 0-~0.3. Do the authors have any speculation what might be driving this? It seems this could lead to an important misclassification of “strangers” to “distant relatives” and perhaps “close relatives”. Apologies if this is just my misinterpretation of the plot (admittedly it is a bit hard to see – maybe adding an alpha transparency to the points may help)

Response: We apologize for difficulties in viewing the points on the scatter plot and have now added histograms displaying the microhaplotype and genomic IBD distributions (see revised **Supplementary Figure 3**). We have also added details to the legend of the figure describing this observation as follows (lines 905-910): *“The microhaplotypes consistently slightly overestimate IBD relative to WGS data at relatedness values less than 0.2. This could be related to intrinsic properties of the hmmIBD algorithm, which was not optimally designed for use with microhaplotype data in its initial iteration. However, this apparent overestimation artifact would not be of particular concern for recurrence characterization,*

where the primary question is whether parasite pairs are identical ($r=1$), siblings ($r=0.5$), or even possibly half-siblings ($r=0.25$). This analysis highlights the need for dedicated methods and tools development for effectively using microhaplotype markers from the wider community.”

- Supp Fig 4: I think there is an error in interpretation or an oversight with respect to the cross-border transmission networks. They occur between Cambodia and Vietnam, not Thailand. Also, there is repeated text in the legend.

Response: Thank you for identifying these errors, we have revised the text accordingly.

REVIEWERS' COMMENTS

Reviewer #2 (Remarks to the Author):

The manuscript by Siegel et al is much improved over the original submission, with inaccuracies and unclear statements removed or clarified. The authors have also added analysis of additional genomes to strengthen their findings. The authors have gone to some length to address all of the reviewers' comments, which is most likely why the revised manuscript has taken many months to reappear, and they should be commended for their perseverance.

Reviewer #3 (Remarks to the Author):

This is a review of the revised manuscript submitted by Siegel et al to Nature Communications, now entitled "Lineage-informative microhaplotypes for recurrence classification and spatio-temporal surveillance of Plasmodium vivax malaria parasites".

I appreciate the time and effort by the authors to address the points raised by me and the other reviewers. The authors have carried out extensive updates to the description of the analytical pipeline and microhaplotype discovery framework and the accompanying github repo to ensure reproducibility and transparency regarding the selection process of their microhap panels. Importantly, flexibility for future extensions/optimization based on particular use cases by others is facilitated through the availability of easy-to-use jupyter notebooks and incorporation into the cloud-based Pv4 data package. I believe this very nice series of notebook resources, the improved description of the framework, and the additional exploration of larger microhaplotype panels adds a lot of robustness to the work. I commend their efforts – the authors have adequately addressed my major comments in this regard. A minor suggestion is to include the github pages link (<https://svsiegel.github.io/vivax-mhaps/>) somewhere in the github repository, I suggest in the repo description website link, as well as in the manuscript. This was really nice! as it holds all the relevant info for those who may not be familiar with github repo directories, folder structure, etc. And I only noticed that this link existed when reviewing the Code and Software checklist.

I also praise the addition of the validation of non-Pv4 samples in response to reviewer 2, I believe it also adds robustness to the overall manuscript and nicely shows the applicability of this panel in less well-sampled regions that are not currently represented in the Pv4 dataset. This was encouraging to see.

All of my minor comments were adequately addressed. And an additional minor comment I noticed – line 299, Figure 7 should be referenced instead of Figure 6.

Overall, I am satisfied that the authors' have adequately addressed my comments, and I thank them for the opportunity to review this excellent manuscript, which I believe is much stronger in its revised form. Congratulations on this impressive body of work!

Reviewer #3 (Remarks on code availability):

The authors have made substantial improvements to the GitHub repository and available code. The README has now been updated and the provided notebooks 1 and 2 are thoroughly described with respect to the Pv4 data curation, marker discovery framework including two separate algorithms for marker selection, example code for panel evaluation for IBD estimation using the R package paneljudge, with a nice addition of exploratory code for users who may wish to customize their own panels.